# Multiply Robust Federated Estimation of Targeted Average Treatment Effects

**Larry Han**
Department of Health Sciences
Northeastern University
Boston, MA 02115
lar.han@northeastern.edu

**Zhu Shen**
Department of Biostatistics
Harvard University
Boston, MA 02115
zhushen@g.harvard.edu

**José R. Zubizarreta**
Departments of Health Care Policy, Biostatistics, and Statistics
Harvard University
Boston, MA 02115
zubizarreta@hcp.med.harvard.edu

## Abstract

Federated or multi-site studies have distinct advantages over single-site studies, including increased generalizability, the ability to study underrepresented populations, and the opportunity to study rare exposures and outcomes. However, these studies are complicated by the need to preserve the privacy of each individual's data, heterogeneity in their covariate distributions, and different data structures between sites. We propose a novel federated approach to derive valid causal inferences for a target population using multi-site data. We adjust for covariate shift and accommodate covariate mismatch between sites by developing a multiply-robust and privacy-preserving nuisance function estimation approach. Our methodology incorporates transfer learning to estimate ensemble weights to combine information from source sites. We show that these learned weights are efficient and optimal under different scenarios. We showcase the finite sample advantages of our approach in terms of efficiency and robustness compared to existing state-of-the-art approaches. We apply our approach to study the treatment effect of percutaneous coronary intervention (PCI) on the duration of hospitalization for patients experiencing acute myocardial infarction (AMI) with data from the Centers for Medicare & Medicaid Services (CMS).

## 1 Introduction

Compared to single-site studies, federated or multi-site studies confer distinct advantages, such as the potential for increased generalizability of findings, the opportunity to learn about underrepresented populations, and the ability to study rare exposures and outcomes. However, deriving valid causal inferences using multi-site data is difficult due to numerous real-world challenges, including heterogeneity of site populations, different data structures, and privacy-preserving constraints stemming from policies such as the General Data Protection Regulation (GDPR) and Health Insurance Portability and Accountability Act (HIPAA) that prohibit direct data pooling.

Recent methodological developments have focused on privacy-preserving estimation strategies. These strategies typically involve sharing summary-level information from multiple data sources [28, 29, 13, 14, 20]. However, they often require restrictive assumptions such as homogeneous data

37th Conference on Neural Information Processing Systems (NeurIPS 2023).

structures and model specifications (e.g., a common set of observed covariates measured using a common data model), which are not realistic in practice.

To address these methodological gaps, we propose a multiply robust and privacy-preserving estimator that leverages multi-site information to estimate causal effects in a target population of interest. Compared to existing approaches, our method allows investigators from different sites to incorporate site-specific covariate information and domain knowledge and provides increased protection against model misspecification. Our method allows for flexible identification under different settings, including systematically missing covariates and different site-specific covariates (termed covariate mismatch). Our proposed method adopts an adaptive ensembling approach that optimally combines estimates from source sites and serves as a data-driven metric for the transportability of source sites. Moreover, the proposed method relaxes the assumption of homogeneous model specifications by adopting a class of multiply robust estimators for estimating the nuisance functions.

## 1.1 Related Work and Contributions

The current literature on multi-site causal inference typically assumes that a common set of confounders is observed in all sites [8, 7, 13, 14]. However, this assumption is rarely met due to variations in local practices, e.g., differing data collection standards and coding practices. In particular, the target site often lacks data on certain covariates available in the source sites, and ignoring them can result in biased and inefficient inference [32]. Recently, [30] proposed a method to address covariate mismatch by integrating source samples with unmeasured confounders and a target sample containing information about these confounders. However, they assumed that the target and source samples are obtained from the same population. [12] extended the method of [30] to the setting where the average treatment effect (ATE) is not identifiable in some sites by constructing control variates. However, their approach is focused on addressing selection biases where preferential selection of units solely depends on the binary outcome, and it is not obvious how one could extend the method to other settings. [32] extended the framework by [8, 7] to handle covariate mismatch by regressing predicted conditional outcomes on effect modifiers and then taking the means of these regression models evaluated on target site samples. Our work leverages ideas from [32] to a more general multi-site federated data setting by utilizing an adaptive weighting approach that optimally combines estimates from source sites.

Most existing approaches in the generalizability and transportability literature deal with heterogeneous covariate distributions by modeling the site selection processes [6, 2, 27, 8, 7]. These approaches, which incorporate inverse probability of selection weights [6, 2], stratification [27], and augmentation [8, 7], involve pooling individual-level information across sites. Our work differs from those in that we preserve individual data privacy, sharing only summary-level information from the target site. Specifically, we adopt density ratio models [24, 9] that only share covariate moments of the target samples. Under certain specifications, these density ratio models are equivalent to logistic regression-based selection models for adjusting heterogeneity between target and source populations. Our approach shares similarities with calibration weighting methods but leverages semi-parametric efficiency theory to enable a closed-form approximation of the variance.

Further, when data sources are heterogeneous, it would be beneficial for investigators at different sites to incorporate site-specific knowledge when specifying candidate models. However, to the best of our knowledge, existing methods require common models to be specified across sites, which may not be realistic or flexible enough [28, 29, 13, 14, 20]. In contrast, this paper takes a different stance; we accommodate variations in outcome and treatment models across sites using a multiply robust estimator, instead of imposing a uniform model. Our work builds on [22], which established an equivalence between doubly robust and multiply robust estimators using mixing weights determined by predictive risks of candidate models [17, 15, 16, 3, 4, 5]. We take advantage of this equivalent form to obtain closed-form expressions for the variance of our federated global estimator. Compared to [32], where they consider a single source site and a single target site, we are able to consider multiple source sites and multiple target sites. To prevent the negative transfer, we combine the estimates from multiple sites via a data-adaptive ensembling approach first proposed in [13].

## 2 Preliminaries

We consider data from $K$ sites, where each site has access to its individual-level data but is prohibited from sharing this data with any other site. The set of sites will be denoted by $\mathcal{K} = \{1, 2, ..., K\}$. We index the target site with $T$ and the source sites as the remaining sites, i.e., $\mathcal{S} = \mathcal{K} \setminus T$.

For each individual $i$, let $Y_i$ denote an observed outcome, which can be continuous or discrete. $X_i \in \mathbb{R}^p$ represents the $p$-dimensional baseline covariates in source site $k \in \mathcal{S}$. $V_i \in \mathbb{R}^q$ represents the (partial) baseline covariates in the target site $T$ such that $V_i \subseteq X_i$. To simplify the presentation, we assume an identical set of covariates across all source sites, although our method can accommodate scenarios where distinct covariate sets are present among the source sites. Let $A_i$ represent a binary treatment indicator, with $A_i = 1$ denoting treatment and $A_i = 0$ denoting control. $R_i$ is a site indicator with $R_i = k$ if patient $i$ is from the site $k$. We observe $n_T$ target observations, $D_T = \{Y_i, V_i, A_i, R_i = T, 1 \le i \le n_T\}$ and $n_k$ source observations, $D_k = \{Y_i, X_i, A_i, R_i = k, 1 \le i \le n_k\}$ for each $k \in \mathcal{S}$. The total sample size is $N = \sum_{k \in \mathcal{K}} n_k$. Under the potential outcomes framework [23, 25], we denote the counterfactual outcomes under treatment and control as $\{Y_i(1), Y_i(0)\}$, and only one of them is observed: $Y_i = A_i Y_i(1) + (1 - A_i) Y_i(0)$ [26]. The data structure is illustrated in Figure 1.

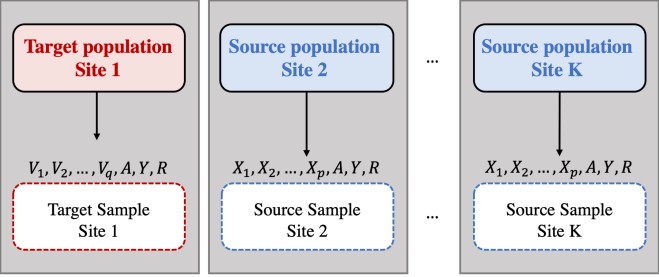

**Figure 1:** Schematic of the data structure in the multi-site setting.

Our goal is to estimate the target average treatment effect (TATE),

$$\Delta_T = \mu_{1,T} - \mu_{0,T} \quad \text{where} \quad \mu_{a,T} = E\{Y_i(a) \mid R_i = T\} \text{ for } a \in \{0, 1\}, \tag{1}$$

where $\mu_{a,T}$ is the mean potential outcome under treatment $a$ in the target population. To identify this quantity, we consider the following assumptions:

(A1) (Consistency): For every individual $i$, if $A_i = a$, then $Y_i = Y_i(a)$.

(A2) (Mean exchangeability over treatment assignment in the target population):
$E\{Y_i(a) \mid V_i = v, A_i, R_i = T\} = E\{Y_i(a) \mid V_i = v, R_i = T\}$.

(A3) (Positivity of treatment assignment in the target population):
$0 < P(A_i = 1 \mid V_i = v, R_i = T) < 1$ for any $v$ s.t. $P(V_i = v \mid R_i = T) > 0$.

(A4) (Mean exchangeability over treatment assignment in the source populations):
$E\{Y_i(a) \mid X_i = x, A_i, R_i = k\} = E\{Y_i(a) \mid X_i = x, R_i = k\}, k \in \mathcal{S}$.

(A5) (Positivity of treatment assignment in the source populations):
$0 < P(A_i = 1 \mid X_i = x, R_i = k) < 1$ for any $x$ s.t. $P(X_i = x \mid R_i = k) > 0, k \in \mathcal{S}$.

(A6) (Mean exchangeability over site selection):
$E\{Y_i(a) \mid V_i = v, R_i = k\} = E\{Y_i(a) \mid V_i = v\}, k \in \mathcal{K}$.

(A7) (Positivity of site selection):
$0 < P(R_i = k \mid V_i = v) < 1$ for $k \in \mathcal{S}$ and any $v$ s.t. $P(V_i = v) > 0$.

Assumption (A1) is the stable unit treatment value assumption (SUTVA), requiring no interference between individuals. Assumption (A2) (Assumption (A4)) states that the mean counterfactual outcome under treatment $a$ is independent of treatment assignment, conditional on baseline covariates in the target (source) populations. For Assumption (A2) and (A4) to hold, we require all effect modifiers to be measured in $V$. Assumption (A3) (Assumption (A5)) states that each individual in the

target (source) populations has a positive probability of receiving each treatment. Assumption (A6) states that the mean counterfactual outcome is independent of site selection, conditional on covariates in the target population. For Assumption (A6) to hold, we require all covariates that are distributed differently between target and source populations (shifted covariates) to be measured in $V$. Thus, if these effect modifiers are measured in $V$, Assumption (A2), (A4) and (A6) automatically hold. Assumption (A7) requires that in each stratum defined by $V$, the probability of being in a source population for each individual is positive. Theorem 1 shows that under Assumption (A1), (A4) - (A7), the mean counterfactual outcome for the target can be identified in the sources.

**Theorem 1.** *If Assumptions (A1) - (A3) hold, the mean counterfactual outcomes in the target population can be identified using the target sample.*

$$\mu_{a,T} = E\{Y_i(a) \mid R_i = T\} = E\{E\{Y_i \mid V_i = v, A_i = a, R_i = T\} \mid R_i = T\}. \qquad (2)$$

*If Assumptions (A1), (A4) - (A7) hold, the mean counterfactual outcomes in the target population can be identified using the source samples.*

$$\begin{aligned} \mu_{a,T} &= E\{Y_i(a) \mid R_i = T\} \\ &= E\{E\{E\{Y_i \mid X_i = x, A_i = a, R_i = k\} \mid V_i = v, R_i = k\} \mid R_i = T\}. \qquad (3) \end{aligned}$$

## 3 Site-specific Estimation

For the target site $k = \{T\}$, a standard augmented inverse propensity weighted (AIPW) estimator is used for $\mu_{a,T}$ as follows

$$\widehat{\mu}_{a,T} = \frac{1}{n_T} \sum_{i=1}^{n} \left[ \frac{I(A_i = a, R_i = T)}{\widehat{\pi}_{a,T}(V_i)} \left\{ Y_i - \widehat{m}_{a,T}(V_i) \right\} + \widehat{m}_{a,T}(V_i) \right], \qquad (4)$$

where $\widehat{m}_{a,T}(V_i)$ is an estimator for $E\{Y_i \mid V_i = v, A_i = a, R_i = T\}$, the outcome model in the target population, and $\widehat{\pi}_{a,T}(V_i)$ is an estimator for $P(A_i = 1 \mid V_i = v, R_i = T)$, the probability of receiving treatment $a$ in the target population.

For each source site $k \in \mathcal{S}$, we propose an estimator for $\mu_{a,T}$ as follows

$$\begin{aligned} \widehat{\mu}_{a,k} = &\frac{1}{n_k} \sum_{i=1}^{n} \left[ \frac{I(A_i = a, R_i = k)}{\widehat{\pi}_{a,k}(X_i)} \widehat{\zeta}_k(V_i) \left\{ Y_i - \widehat{m}_{a,k}(X_i) \right\} \right] \\ &+ \frac{1}{n_k} \sum_{i=1}^{n} \left[ I(R_i = k) \widehat{\zeta}_k(V_i) \left\{ \widehat{m}_{a,k}(X_i) - \widehat{\tau}_{a,k}(V_i) \right\} \right] + \frac{1}{n_T} \sum_{i=1}^{n} I(R_i = T) \widehat{\tau}_{a,k}(V_i), \qquad (5) \end{aligned}$$

where $\widehat{\tau}_{a,k}(V_i)$ is an estimator for $E\{m_{a,k}(x) \mid V_i = v, R_i = k\}$ and $\widehat{m}_{a,k}(X_i)$ is an estimator for $E\{Y_i \mid X_i = x, A_i = a, R_i = k\}$. $\widehat{\zeta}_k(V_i)$ estimates $f(V_i \mid R_i = T)/f(V_i \mid R_i = k)$, the density ratios of covariate distributions in the target population $T$ and source population $k \in \mathcal{S}$. $\widehat{\pi}_{a,k}(X_i)$ estimates $P(A_i = 1 \mid X_i = x, R_i = k)$, the probability of receiving treatment $a$ in source $k \in \mathcal{S}$. The estimators (4) and (5) are derived leveraging semi-nonparametric theory. They can be seen as a functional Taylor expansion of the target functional $\mu_{a,T}$. This method is commonly used in causal inference to correct for the bias of the plug-in estimators. These resulting estimators exhibit favorable statistical properties, even when slow-converging non-parametric techniques are used for nuisance estimation. For a more comprehensive discussion, interested readers can refer to [21] and [18].

Compared to the transportation estimators in [8, 7], we introduce two additional nuisance functions, $\zeta_k(V_i)$ and $\tau_{a,k}(V_i)$. Specifically, $\zeta_k(V_i)$ accounts for covariate shift across sites while preserving individual's data privacy, while $\tau_{a,k}(V_i)$ is introduced to address covariate mismatch across sites. We provide estimation procedures for these nuisance functions in the following subsections, and the theoretical guarantees of the estimator are presented in Section 5.

### 3.1 Density Ratio Weighting

Most existing methods adjust for covariate shift across sites by relying on inverse probability of selection weighting, which requires pooling target and source samples. However, such pooling is

often not possible due to data privacy regulations. We consider a density ratio weighting approach, which offers equivalent estimation without the need for direct data pooling (see Appendix A).

Formally, as in [13], we model the density ratios of covariate distributions in the target $T$ and source $k \in \mathcal{S}$ by specifying an exponential tilt model [24, 9]; $\zeta_k(V_i; \theta_k) = f(V_i \mid R_i = T)/f(V_i \mid R_i = k) = \exp\left\{-\theta_k^\top \psi(V_i)\right\}$ where $f(V_i \mid R_i = T)$ and $f(V_i \mid R_i = k)$ are density functions of covariates $V_i$ in the target $T$ and source $k \in \mathcal{S}$, respectively, and $\psi(V_i)$ is some $d$-dimensional basis with 1 as its first element. With this formulation, $\zeta_k(V_i; \theta_k) = 1$ for $\theta_k = 0$ and $\int \zeta_k(V_i; \theta_k) f(V_i \mid R_i = k) dx = 1$. If we choose $\psi(V_i) = V_i$, we can recover the entire class of natural exponential family distributions. If we include higher-order terms, the exponential tilt model has greater flexibility in characterizing the heterogeneity between two populations [10]. We solve for $\widehat{\theta}_k$ with the following estimating equation

$$\frac{1}{n_T} \sum_{i=1}^{N} I\left(R_i = T\right) \psi\left(V_i\right) = \frac{1}{n_k} \sum_{i=1}^{N} I\left(R_i = k\right) \psi\left(V_i\right) \exp\left\{-\theta_k^\top \psi(V_i)\right\}. \tag{6}$$

This procedure preserves individual privacy; choosing $\psi(V_i) = V_i$, the target site only needs to share its covariate means with the source sites; each source site then solves (6) with its own data to obtain the density ratios.

## 3.2  Multiply Robust Estimation

We relax the assumption of homogeneous model specifications across sites and allow each site to propose multiple models for nuisance functions. Our proposal follows the construction of multiply robust estimators for nuisance functions via a model-mixing approach [22].

Formally, for each site $k \in \mathcal{K}$, we consider a set of $J$ candidate treatment models for the propensity scores $\{\pi_{a,k}^j(x) : j \in \mathcal{J} = \{1, ..., J\}\}$. Let $\widehat{\pi}_{a,k}^j(x)$ be the estimator of $\pi_{a,k}^j(x)$ obtained by fitting the corresponding candidate models on the data, which can be parametric, semiparametric, or nonparametric machine learning models. $\widehat{\pi}_{a,k}(X_i) = \sum_{j=1}^{J} \widehat{\Lambda}_j \widehat{\pi}_{a,k}^j(X_i)$ denotes the weighted predictions of propensity scores, with weights $\widehat{\Lambda}_j$ assigned to predictions by each candidate model $j$. To calculate the weights $\widehat{\Lambda}_j$, we adapt a model-mixing algorithm developed in [31] and [22] based on the cumulative predictive risks of candidate models.

First, we randomly partition the data within each site into a training set $D_k^{\text{train}}$ of units indexed by $\{1, ..., n_k^{\text{train}}\}$ and a validation set $D_k^{\text{val}}$ of units indexed by $\{n_k^{\text{train}} + 1, ..., n_k\}$. Then, each candidate treatment model is fit on $D_k^{\text{train}}$ to obtain $\widehat{\pi}_{a,n_k^{\text{train}}}^j$ for $j \in \mathcal{J}$. The model-mixing weights are determined by the models' predictive risks assessed on $D_k^{\text{val}}$ according to the Bernoulli likelihood. Specifically,

$$\widehat{\Lambda}_j = \left(n_k - n_k^{\text{train}}\right)^{-1} \sum_{i=n_k^{\text{train}}+1}^{n_k} \widehat{\Lambda}_{j,i} \quad \text{and}$$

$$\widehat{\Lambda}_{j,i} = \frac{\Pi_{q=n_k^{\text{train}}+1}^{i-1} \widehat{\pi}_{a,n_k^{\text{train}}}^j (X_q)^{A_q} \left\{1 - \widehat{\pi}_{a,n_k^{\text{train}}}^j (X_q)\right\}^{1-A_q}}{\sum_{j'=1}^{J} \Pi_{q=n_k^{\text{train}}+1}^{i-1} \widehat{\pi}_{a,n_k^{\text{train}}}^{j'} (X_q)^{A_q} \left\{1 - \widehat{\pi}_{a,n_k^{\text{train}}}^{j'} (X_q)\right\}^{1-A_q}} \quad \text{for} \quad n_k^{\text{train}} + 2 \leq i \leq n_k,$$
$$\tag{7}$$

where $\widehat{\Lambda}_{j,n_k^{\text{train}}+1} = 1/J$. The model-mixing estimators are consistent if one of the $j \in \mathcal{J}$ candidate models is correctly specified [22]. A similar strategy can be used for conditional outcomes $m_{a,k}(X_i)$ by combining a set of $L$ candidate outcome models $\{m_{a,k}^l(x) : l \in \mathcal{L} = \{1, ..., L\}\}$. We obtain $\widehat{m}_{a,k}(X_i) = \sum_{l=1}^{L} \widehat{\Omega}_l \widehat{m}_{a,k}^l(X_i)$ as the predicted outcomes with weights $\widehat{\Omega}_l$ of candidate outcomes models under treatment $a$ in site $k$. Further details are provided in Appendix B.

## 3.3  Covariate Mismatch

To account for covariate mismatch, we adapt the approach in [32], introducing the nuisance function $\tau_{a,k}(V_i) = E\{m_{a,k}(x) \mid V_i = v, R_i = k\}$, where $m_{a,k}(x)$ is the outcome regression for treatment $a$

in site $k$. First, we estimate $m_{a,k}(X_i)$ by regressing the outcome $Y_i$ on covariates $X_i$ among units receiving treatment $a$ in site $k$. We then regress $\widehat{m}_{a,k}(X_i)$, the estimates from the previous step, on $V_i$ in the source site $k$ to obtain $\widehat{\tau}_{a,k}(x)$. By doing so, we project all site-specific estimates of conditional outcomes to a common hyperplane defined by $V_i$. If all effect modifiers that are distributed differently between target and source populations are measured in $V_i$, then the information contained in the projected site-specific estimates can be transported to the target site. Finally, we take the mean of $\widehat{\tau}_{a,k}(x)$ over the target sample, which gives us the transported estimate $\hat{\tau}_{a,k}(V_i)$ for the mean counterfactual outcomes under treatment $a$ in the target population.

## 4  Federated Estimation

Let $\widehat{\mu}_{a,T}$ denote the estimate of $\mu_{a,T}$ based on target data only and $\widehat{\mu}_{a,k}$ be the estimates of $\mu_{a,T}$ using source data $k \in \mathcal{S}$. We propose a general form of the federated global estimator as follows

$$\widehat{\mu}_{a,G} = \widehat{\mu}_{a,T} + \sum_{k \in \mathcal{K}} \widehat{\eta}_k \left\{ \widehat{\mu}_{a,k} - \widehat{\mu}_{a,T} \right\}, \tag{8}$$

where $\widehat{\eta}_k \geq 0$ is a non-negative weight assigned to site-specific estimates and $\sum_{k \in \mathcal{K}} \widehat{\eta}_k = 1$. The role of $\eta_k$ is to determine the ensemble weight given to the site-specific estimates. We can employ diverse weighting methods by selecting appropriate values of $\eta_k$. For example, if $\eta_k = 0$, the global estimator is simply the estimator based on target data only; if $\eta_k = n_k/N$, the global estimator combines site-specific estimates by their sample sizes; if $\eta_k = (1/\sigma_k^2)/\sum_{j \in \mathcal{K}}(1/\sigma_j^2)$ where $\sigma_k^2 = \text{Var}(\widehat{\mu}_{a,k})$, the global estimator is the inverse variance weighting estimator, which is known to be appropriate when working models are homogeneous across sites [29].

In transportability studies, preventing negative transfer is critical when there are multiple, potentially biased source sites. If source sites are biased, both the sample size-weighted estimator and the variance-weighted estimator would inherit this bias. By examining the MSE of the data-adaptive estimator to the limiting estimand of the target estimator, the MSE can be decomposed into a variance term that can be minimized by regression of influence functions obtained from the asymptotic linear expansion of the target and source estimates, and an asymptotic bias term. This allows us to rewrite the problem of solving for ensemble weights as an adaptive LASSO problem [13, 14] as follows:

$$\widehat{\eta}_{k,L_1} = \arg\min_{\eta_k \geq 0} \sum_{i=1}^{N} \left[ \widehat{\xi}_{T,i}(a) - \sum_{k \in \mathcal{K}} \eta_k \left( \widehat{\xi}_{T,i}(a) - \widehat{\xi}_{k,i}(a) - \widehat{\delta}_k \right) \right]^2 + \lambda \sum_{k \in \mathcal{K}} |\eta_k| \, \widehat{\delta}_k^2, \tag{9}$$

where $\eta_{k,L_1}$ denotes the data-adaptive weights; $\widehat{\xi}_{T,i}(a)$ and $\widehat{\xi}_{k,i}(a)$ are the estimated influence functions for the target and source site estimators (see Appendix E.3 for the exact form of the influence functions). The estimated difference $\widehat{\delta}_k = \widehat{\mu}_{a,k} - \widehat{\mu}_{a,T}$ quantifies the bias between the estimate from source $k \in \mathcal{S}$ and the estimate from the target $T$. The tuning parameter $\lambda$ determines the penalty imposed on source site estimates and in practice, is chosen via cross-validation. Specifically, we create a grid of values of $\lambda$ and iteratively train and evaluate the model using different $\lambda$ values, selecting the one with the lowest average validation error after multiple sample splits.

We estimate the variance of $\widehat{\mu}_{a,G}$ using the estimated influence functions for $\widehat{\mu}_{a,T}$ and $\widehat{\mu}_{a,k}$. By the central limit theorem, $\sqrt{N}(\widehat{\mu}_{a,G} - \bar{\mu}_{a,G}) \xrightarrow{d} \mathcal{N}(0, \Sigma)$, where $\Sigma = E\{\sum_{k \in \mathcal{K}} \bar{\eta}_k \xi_{k,i}(a)\}^2$ and $\bar{\mu}_{a,G}$ and $\bar{\eta}_k$ denote the limiting values of $\widehat{\mu}_{a,G}$ and $\widehat{\eta}_k$ respectively. The standard error of $\widehat{\mu}_{a,G}$ is estimated as $\sqrt{\widehat{\Sigma}/N}$ where $\widehat{\Sigma} = N^{-1} \sum_{k \in \mathcal{K}} \sum_{i=1}^{n_k} \{\widehat{\eta}_k \widehat{\xi}_{k,i}(a)\}^2$. A two-sided $(1-\alpha) \times 100\%$ confidence interval for $\mu_{a,G}$ is

$$\widehat{\mathcal{C}}_\alpha = \left[ \widehat{\mu}_{a,G} - \sqrt{\widehat{\Sigma}/N} \mathcal{Z}_{\alpha/2}, \quad \widehat{\mu}_{a,G} + \sqrt{\widehat{\Sigma}/N} \mathcal{Z}_{\alpha/2} \right], \tag{10}$$

where $\mathcal{Z}_{\alpha/2}$ is the $1 - \alpha/2$ quantile for a standard normal distribution.

## 5  Theoretical Guarantees

In this section, we first establish the theoretical properties of the site-specific estimators constructed with the multiply robust model-mixing approach. Define $\bar{\pi}_{a,k}^j$, $\bar{m}_{a,k}^l$, $\bar{\tau}_{a,k}$ and $\bar{\zeta}_k$ as non-stochastic

functionals that the corresponding estimators $\widehat{\pi}^j_{a,k}$, $\widehat{m}^l_{a,k}$, $\widehat{\tau}_{a,k}$ and $\widehat{\zeta}_k$ converge to for $k \in \mathcal{K}^1$. That is,

$$\|\widehat{\pi}^j_{a,k} - \overline{\pi}^j_{a,k}\| = o_p(1), \quad \|\widehat{m}^l_{a,k} - \overline{m}^l_{a,k}\| = o_p(1), \quad \|\widehat{\tau}_{a,k} - \overline{\tau}_{a,k}\| = o_p(1), \quad \|\widehat{\zeta}_k - \overline{\zeta}_k\| = o_p(1).$$

As shown in Lemmas E.1 and E.2 in Appendix E.2, the $L_2$ risks of the model-mixing estimators $\widehat{\pi}_{a,k}$ and $\widehat{m}_{a,k}$ are bounded by the smallest risks of all candidate models plus a remainder term that vanishes at a faster rate than the risks themselves. Leveraging Lemmas E.1 and E.2, we show the consistency and asymptotic normality of the site-specific estimators in Theorem 2. The proofs are given in the Appendix E.4.

**Theorem 2.** *Suppose that the conditions in Lemmas E.1 and E.2 hold, and for $k \in \mathcal{K}$ that $\widehat{\pi}^j_{a,k}$, $\widehat{m}^l_{a,k}$, $\widehat{\zeta}_k$, $\widehat{\tau}_{a,k}$, $\overline{\pi}^j_{a,k}$, $\overline{m}^l_{a,k}$, $\overline{\zeta}_k$ and $\overline{\tau}_{a,k}$ are uniformly bounded for all treatment models $j \in \mathcal{J}$ and for all outcome models $l \in \mathcal{L}$. Consider the following conditions:*

> *(B1)* $\overline{\pi}^j_{a,k} = \pi_{a,k}$ *for some* $j \in \mathcal{J}$,
> *(B2)* $\overline{m}^l_{a,k} = m_{a,k}$ *for some* $l \in \mathcal{L}$,
>
> *(C1)* $\overline{\zeta}_k = \zeta_k$,
> *(C2)* $\overline{\tau}_{a,k} = \tau_{a,k}$.

*Then, under Assumptions (A1) - (A7), and if one of (B1) or (B2) and one of (C1) or (C2) hold,*

$$\|\widehat{\mu}_{a,k} - \mu_{a,T}\| = O_p \left( n^{-1/2} + \|\widehat{\pi}_{a,k} - \pi_{a,k}\|\|\widehat{m}_{a,k} - m_{a,k}\| + \|\widehat{\zeta}_k - \zeta_k\|\|\widehat{\tau}_{a,k} - \tau_{a,k}\| \right). \quad (11)$$

*Further, if the nuisance estimators satisfy the following convergence rate*

$$\|\widehat{m}_{a,k} - m_{a,k}\| \, \|\widehat{\pi}_{a,k} - \pi_{a,k}\| = o_p(1/\sqrt{n}), \quad \|\widehat{\zeta}_k - \zeta_k\| \, \|\widehat{\tau}_{a,k} - \tau_{a,k}\| = o_p(1/\sqrt{n}), \quad (12)$$

*then $\sqrt{n}(\widehat{\mu}_{a,k} - \mu_{a,T})$ asymptotically converges to a normal distribution with mean zero and asymptotic variance equal to the semiparametric efficiency bound.*

In Theorem 3, we establish the consistency and asymptotic normality of the federated global estimator. These properties are attained without requiring consistency in the site-specific estimators from the source sites, a consequence of the adaptive weighting method employed. We further delve into conditions in which leveraging information from source sites can enhance the efficiency of the estimator constructed solely with the target data. The proofs are given in the Appendix E.5.

**Theorem 3.** *Under Assumptions (A1) - (A7) and the regularity conditions specified in the Appendix E.5, the federated global estimator of $\Delta_T$, given by $\widehat{\Delta}_G = \widehat{\mu}_{1,G} - \widehat{\mu}_{0,G}$, is consistent and asymptotically normal,*

$$\sqrt{N/\widehat{\mathcal{V}}} \left( \widehat{\Delta}_G - \Delta_T \right) \xrightarrow{d} \mathcal{N}(0,1), \quad (13)$$

*with the variance estimated consistently as $\widehat{\mathcal{V}}$. The variance of $\widehat{\Delta}_G$ is no larger than that of the estimator based on target data only, $\widehat{\Delta}_T = \widehat{\mu}_{1,T} - \widehat{\mu}_{0,T}$. Further, if there exist some source sites with consistent estimators of $\Delta_T$ and satisfy conditions specified in the Appendix E.5, the variance of $\widehat{\Delta}_G$ is strictly smaller than $\widehat{\Delta}_T$.*

## 6 Numerical Experiments

We evaluate the finite sample properties of five different estimators: (i) an augmented inverse probability weighted (AIPW) estimator using data from the target site only (Target), (ii) an AIPW estimator that weights each site proportionally to its sample size (SS), (iii) an AIPW estimator that weights each site inverse-proportionally to its variance (IVW), (iv) an AIPW estimator that weights each site with the $L_1$ weights defined in (9) (AIPW-$L_1$), and (v) a multiply robust estimator with the

---

[1]When $k = T$, the site-specific estimator takes the form in (4) and we only specify the outcome and treatment models. To establish the consistency and asymptotic normality of the target site-specific estimator, we require one of the Assumptions (B1) and (B2) in Theorem 3 to hold. Then (11) is simplified since we only need to consider the convergence rate of $\|\widehat{m}_{a,k} - m_{a,k}\| \, \|\widehat{\pi}_{a,k} - \pi_{a,k}\|$.

$L_1$ weights defined in (9) (MR-$L_1$). Across different settings, we examine the performance of each estimator in terms of mean absolute error, root mean square error, and coverage and length of 95% confidence intervals (CI) across 500 simulations.

We consider a total of five sites and fix the first site as the target site with a relatively small sample size of 300. The source sites have larger sample sizes of $\{500, 500, 1000, 1000\}$. We model heterogeneity in the covariate distributions across sites with skewed normal distributions and varying levels of skewness in each site, $X_{kp} \sim \mathcal{SN}\left(x; \Xi_{kp}, \Omega_{kp}^2, \gamma_{kp}\right)$, where $k \in \{1, ..., 5\}$ indexes each site and $p \in \{1, ..., 4\}$ indexes the covariates; $\Xi_{kp}, \Omega_{kp}^2$ and $\gamma_{kp}$ are the location, scale, and skewness parameters, respectively. Following [19], we also generate covariates $Z_{kp}$ as non-linear transformation of $X_{kp}$ such that $Z_{k1} = \exp(X_{k1}/2)$, $Z_{k2} = X_{k2}/\{1+\exp(X_{k1})\}+10$, $Z_{k3} = (X_{k1}X_{k3}/25+0.6)^3$ and $Z_{k4} = (X_{k2} + X_{k4} + 20)^2$.

To demonstrate that our proposed MR-$L_1$ estimator can handle covariate mismatch across sites, we consider the setting where there exists covariate mismatch across sites, i.e. $V_k = \{X_{k1}, X_{k2}\}$ for the target site and $X_k = \{X_{k1}, X_{k2}, X_{k3}, X_{k4}\}$ for the source sites. In Appendix D.1, we also provide results for the setting where there is no covariate mismatch between sites. Specifically, we generate potential outcomes as

$$Y_k(a) = 210 + X_k\beta_x + \varepsilon_k \tag{14}$$

where $\beta_x = (27.4, 13.7, 0, 0)$ for units in the target site and $\beta_x = (27.4, 13.7, 13.7, 13.7)$ for units in the source sites. Similarly, the treatment is generated as

$$A_k \sim \text{Bernoulli}(\pi_k) \quad \pi_k = \text{expit}(X_k\alpha_x) \tag{15}$$

where $\alpha_x = (-1, 0.5, 0, 0)$ for units in the target site and $\alpha_x = (-1, 0.5, -0.25, -0.1)$ for units in the source site. With this data generation scheme, the true TATE is $\Delta_T = 0$.

The AIPW-$L_1$ estimator, which requires common models across sites, only uses the shared covariates ($X_{k1}$ and $X_{k2}$) to specify outcome and treatment models for all sites. On the other hand, our MR-$L_1$ estimator allows for different covariates in different sites, so we utilize both shared covariates with the target site and unique covariates to specify the outcome and treatment models in the source sites.

For the site-specific estimators based on MR-$L_1$, we adaptively mix two outcome models and two treatment models. In particular, we specify the first model with the covariates $X_{kp}$, and the second model with the covariates $Z_{kp}$.

**Table 1:** Mean absolute error (MAE), root mean squared error (RMSE), coverage (Cov.), and length (Len.) of 95% CIs based on 500 simulated data sets in covariate mismatch settings.

|  | Target | SS | IVW | AIPW-$L_1$ | MR-$L_1$ |
|---|---|---|---|---|---|
| MAE | 0.108 | 4.331 | 0.150 | 0.107 | 0.053 |
| RMSE | 0.136 | 4.401 | 0.186 | 0.134 | 0.067 |
| Cov. | 0.946 | 1.000 | 0.882 | 0.950 | 0.944 |
| Len. | 0.538 | 26.024 | 0.553 | 0.536 | 0.253 |

In Table 1, we observe that the AIPW-$L_1$ estimator exhibits similar MAE, RMSE, coverage, and length of confidence intervals as the Target estimator while outperforming the SS and IVW estimators. This is because relying solely on shared covariates leads to significant biases in all source sites (Figure 2, left panel), and the AIPW-$L_1$ estimator assigns nearly all of the ensemble weight to the target site so as to reduce bias.

In contrast, the MR-$L_1$ estimator outperforms the AIPW-$L_1$ estimator by exhibiting substantially smaller MAE, lower RMSE, and better coverage. This improvement can be attributed to the inclusion of unique covariates from the source sites, which allows for the recovery of true models in those sites and contributes to accurate estimation of $\Delta_T$ (Figure 2, right panel). These findings suggest that neglecting covariate mismatch by solely relying on shared covariates can lead to highly biased results.

## 7 Case Study

We employ our approach to investigate the treatment effect of percutaneous coronary intervention (PCI) on the duration of hospitalization for individuals experiencing acute myocardial infarction

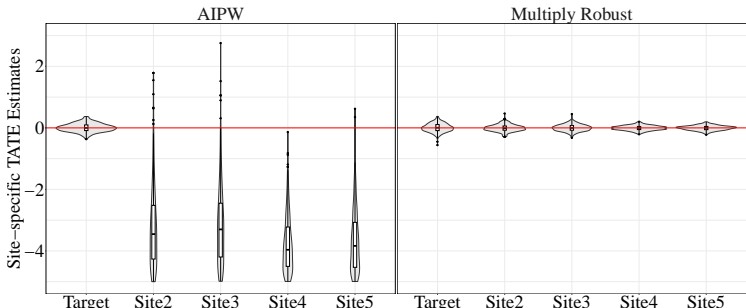

**Figure 2:** Estimates of the TATE based on 500 simulated data sets with covariate mismatch comparing the site-specific estimators with nuisance functions estimated by AIPW and by multiply robust model-mixing.

(AMI), one of the leading causes of hospitalization and mortality in the United States. We utilize a dataset from the Centers for Medicare & Medicaid Services (CMS), comprising a representative cohort of Medicare beneficiaries admitted to short-term acute-care hospitals. We select Maine as our target state of interest and incorporate data from the other 48 continental states to augment the Maine-specific treatment effect estimation.

We choose Maine as our target state for studying PCI treatment quality due to its limited AMI patient population. This scarcity of data in Maine necessitates the integration of information from other states for more precise treatment quality assessment. We consider a scenario where we only possess basic patient demographic information, including age, race, gender, and principal diagnosis categories in Maine. In contrast, the other source states have access to more extensive patient medical histories and have collected additional covariates, namely patient comorbidities.

The AIPW-$L_1$ estimator requires a homogeneous model across states, so we define outcome and treatment models within each state using common patient demographic variables. By contrast, for our proposed MR-$L_1$ estimator, we use patient demographic variables for outcome and treatment models in Maine, while in other states, we use both patient demographic and comorbidity variables for model specification. We adaptively combine two outcome models and two treatment models in the source states. The first model is defined using patient demographic variables, and the second model is defined with both patient demographic and comorbidity variables.

The estimation results of the case study are illustrated in Figure 3 using the five estimators. We observe that the AIPW-$L_1$ and our proposed MR-$L_1$ estimators yield point estimates that are close to the estimate by the Target estimator, while SS and IVW estimators exhibit substantial bias. The estimated treatment effect of PCI is approximately -7.6 days, indicating that if patients diagnosed with AMI in Maine had received PCI treatment instead of medical management alone, they would have experienced an estimated reduction in their hospitalization duration of approximately 7.6 days[2]. Both the AIPW-$L_1$ and our proposed MR-$L_1$ estimators exhibit efficiency gain with smaller standard errors compared to the Target estimator.

| Estimator | Est. | (CI) |
|---|---|---|
| Target | −7.63 | (−11.45 −3.81) |
| SS | −9.93 | (−15.29 −4.56) |
| IVW | −8.94 | (−9.47 −8.41) |
| AIPW–L1 | −7.84 | (−9.60 −6.09) |
| MR–L1 | −7.49 | (−9.82 −5.16) |

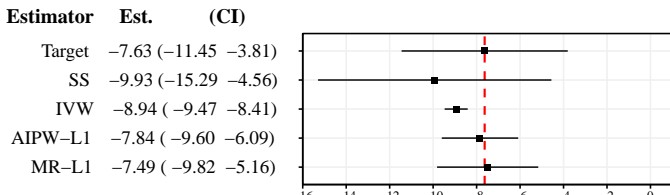

**Figure 3:** Estimates of PCI treatment effect in Maine with covariate mismatch in patient comorbidities

---

[2]We assigned patients who died in the hospital a length of stay equivalent to the maximum of either their actual length of stay or the 99th percentile value of the length of stay in the dataset [1]. This modification is intended to prevent hospitals where patients often die early in their hospitalization from inaccurately appearing more efficient than other hospitals. Using the exact length of stay as the outcome may result in a lower estimated effect of the PCI treatment.

To highlight the ensemble weights corresponding to the source states, we generate Figure 4 illustrating the estimated weights associated with each state. States with positive weights are colored in blue, with darker shades indicating larger weights. States with zero weights are uncolored (white). We observe that both SS and IVW estimators integrate all state-specific estimates, regardless of how dissimilar these estimates are to the target estimate, thereby introducing negative transfer. In contrast, with the proposed adaptive ensemble approach, both AIPW-$L_1$ and MR-$L_1$ estimators only incorporate estimates from a few states whose state-specific estimates are close to the target state estimate, ensuring that the federated estimate closely aligns with the target estimate while achieving an efficiency gain.

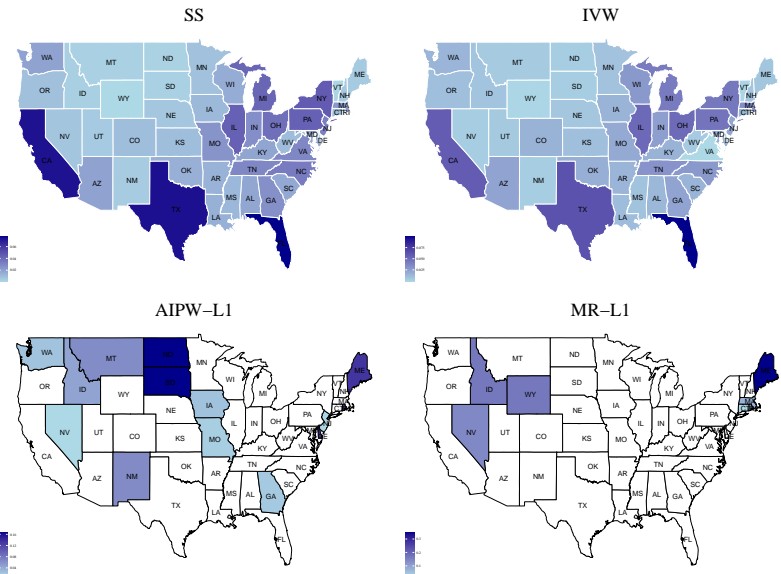

**Figure 4:** Federation weights across states for the PCI treatment effect in Maine using four federated estimators.

# 8 Conclusion

We have proposed a novel federated approach for privacy-preserving, multiply robust, and covariate flexible estimation of causal effects. Compared to existing federated methods, our proposed approach accommodates covariate shift and covariate mismatch across sites, while guaranteeing efficient estimation and preserving privacy in the sense that only covariate means of the target samples are shared in a single round of communication. Our proposal allows investigators in each site to have greater flexibility in specifying candidate models by utilizing site-specific information. Moreover, our method utilizes adaptive ensemble weights to avoid negative transfer in the federation process.

In practical scenarios, the proposed method would be particularly valuable in multi-source research settings (e.g., data consortia) lacking a central data collection point. These methods would be beneficial for combining dissimilar data sources from various hospitals. Additionally, our proposed methodology would find utility in other contexts, such as multi-source clinics, firms, or schools needing to assess their performance relative to peer institutions. These situations may involve data privacy or propriety concerns that prevent the sharing of detailed, unit-level data.

To handle high-dimensional covariates, future research can explore ways to jointly model the propensity score and density ratio to reduce the dimension of parameters for population balancing.

In real-world scenarios, a potential weakness of our proposed approach is the risk of breaching patient privacy. We do not claim or test for formal privacy guarantees of the method; our work is privacy-preserving in the sense that only summary-level information is shared between target and source sites. Any given site is not allowed to specify a specific group and query this group's statistics in any other source site's dataset. Consequently, the target site cannot easily perform membership inference attacks. Nonetheless, for a more formal demonstration of privacy preservation, future research could employ the differential privacy (DP) method [11].

# 9   Acknowledgements

We would like to thank Yige Li, Bijan Niknam, Zhenghao Zeng, Wei Li, feedback received at the 2023 American Causal Inference Conference (ACIC) and the 2023 Institute for Operations Research and the Management Sciences (INFORMS) Annual Meeting, and the five anonymous reviewers for their helpful comments and input that greatly enhanced the paper.

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

# APPENDIX

## A   Equivalence of density ratio weighting and inverse of selection probability weighting

We show that the inverse probability of selection weighting (IPSW) to the site $k$, $\rho_k(V_i) = \frac{1-P(R_i=k|V_i=v)}{P(R_i=k|V_i=v)}$ is equivalent to the density ratio weighting $\zeta_k(V_i) = \frac{P(V_i=v|R_i=T)}{P(V_i=v|R_i=k)}$.

Under Assumptions (A1) - (A7), the site-specific estimators based on the IPSW for $\mu_{a,T}$ is

$$\widehat{\mu}_{a,k} = \frac{1}{n_T} \sum_{i=1}^{n} \left[ \frac{I(A_i = a, R_i = k)}{\widehat{\pi}_{a,k}(X_i)} \widehat{\rho}_k(V_i)\{Y_i - \widehat{m}_{a,k}(X_i)\} \right]$$

$$+ \frac{1}{n_T} \sum_{i=1}^{n} \left[ I(R_i = k)\widehat{\rho}_k(V_i)\left\{\widehat{m}_{a,k}(X_i) - \widehat{\tau}_{a,k}(V_i)\right\} \right]$$

$$+ \frac{1}{n_T} \sum_{i=1}^{n} I(R_i = T)\widehat{\tau}_{a,k}(V_i) \tag{16}$$

where $\widehat{\rho}_k(V_i) = \frac{1-\widehat{P}(R_i=k|V_i=v)}{\widehat{P}(R_i=k|V_i=v)}$ is an estimator for $\rho_k(V_i)$. Applying Baye's rule, we show that the IPSW is equivalent to the density ratio weighting up to a constant,

$$\rho_k(V_i) = \frac{1 - P(R_i = k \mid V_i = v)}{P(R_i = k \mid V_i = v)}$$

$$= \frac{P(R_i = T \mid V_i = v)}{P(R_i = k \mid V_i = v)}$$

$$= \frac{P(V_i = v \mid R_i = T)P(R_i = T)}{P(V_i = v \mid R_i = k)P(R_i = k)}$$

$$= \zeta_k(V_i)\frac{P(R_i = T)}{P(R_i = k)}. \tag{17}$$

We re-write (16) by substituting $\widehat{\rho}_k(V_i)$ with $\widehat{\zeta}_k(V_i)$,

$$\widehat{\mu}_{a,k} = \frac{1}{n_T} \sum_{i=1}^{n} \left[ \frac{I(A_i = a, R_i = k)}{\widehat{\pi}_{a,k}(X_i)} \widehat{\zeta}_k(V_i)\frac{\widehat{P}(R_i = T)}{\widehat{P}(R_i = k)}\{Y_i - \widehat{m}_{a,k}(X_i)\} \right]$$

$$+ \frac{1}{n_T} \sum_{i=1}^{n} \left[ I(R_i = k)\widehat{\zeta}_k(V_i)\frac{\widehat{P}(R_i = T)}{\widehat{P}(R_i = k)}\{\widehat{m}_{a,k}(X_i) - \widehat{\tau}_{a,k}(V_i)\} \right]$$

$$+ \frac{1}{n_T} \sum_{i=1}^{n} I(R_i = T)\widehat{\tau}_{a,k}(V_i). \tag{18}$$

A reasonable estimator $\frac{\widehat{P}(R_i=T)}{\widehat{P}(R_i=k)}$ is $\frac{n_T}{n_k}$. Therefore, we recover our proposed site-specific estimator for $\mu_{a,T}$,

$$\widehat{\mu}_{a,k} = \frac{1}{n_k} \sum_{i=1}^{n} \left[ \frac{I(A_i = a, R_i = k)}{\widehat{\pi}_{a,k}(X_i)} \widehat{\zeta}_k(V_i)\{Y_i - \widehat{m}_{a,k}(X_i)\} \right]$$

$$+ \frac{1}{n_k} \sum_{i=1}^{n} \left[ I(R_i = k)\widehat{\zeta}_k(V_i)\{\widehat{m}_{a,k}(X_i) - \widehat{\tau}_{a,k}(V_i)\} \right]$$

$$+ \frac{1}{n_T} \sum_{i=1}^{n} I(R_i = T)\widehat{\tau}_{a,k}(V_i). \tag{19}$$

## B   Multiply robust estimation for $m_{a,k}$

For multiply robust outcome estimation within each site $k$, we consider a set of $L$ candidate models for conditional outcomes $\{m_{a,k}^l(x) : l \in \mathcal{L} = \{1, ..., L\}\}$. Let $\hat{m}_{a,k}^l(x)$ be the estimates of $m_{a,k}^l(x)$

obtained by fitting the corresponding candidate models, which can be parametric, semiparametric, or nonparametric machine learning models. Let $\widehat{m}_{a,k}(X_i) = \sum_{l=1}^{L} \widehat{\Omega}_l \widehat{m}_{a,k}^l(X_i)$ be the predictions with ensemble weights $\widehat{\Omega}_l$ of candidate outcome models under treatment $a$ in site $k$. We derive the ensemble weights $\widehat{\Omega}_l$ based on the cumulative predictive risk of candidate models. In particular, we denote the data corresponding to treated and control units as $D_{k,1}$ and $D_{k,0}$ respectively, and the sample sizes of $D_{k,1}$ and $D_{k,0}$ are denoted as $n_{k,1}$ and $n_{k,0}$. Consider the treated samples first; we randomly partition $D_{k,1}$ into a training set $D_{k,1}^{\text{train}}$ of units $i \in \{1, ..., n_{k,1}^{\text{train}}\}$ and a validation set $D_{k,1}^{\text{val}}$ of units $i \in \{n_{k,1}^{\text{train}} + 1, ..., n_{k,1}\}$. Then, each candidate outcome model is fit on $D_{k,1}^{\text{train}}$ to obtain $\widehat{m}_{a,n_{k,1}^{\text{train}}}^l$ for $l \in \mathcal{L}$.

If the outcome is binary, the ensemble weights are determined by the fitted models' predictive risks evaluated on $D_{k,1}^{\text{val}}$ according to the Bernoulli likelihood as shown in (7). If the outcome is continuous, the ensemble weights are alternatively determined by the mean squared errors of the fitted models on $D_{k,1}^{\text{val}}$ [15, 4]. Specifically,

$$\widehat{\Omega}_l = \left(n_{k,1} - n_{k,1}^{\text{train}}\right)^{-1} \sum_{i=n_{k,1}^{\text{train}}+1}^{n_{k,1}} \widehat{\Omega}_{l,i} \text{ and}$$

$$\widehat{\Omega}_{l,i} = \frac{\exp\left[-\kappa \sum_{q=n_{k,1}^{\text{train}}+1}^{i-1} \left\{Y_q - \widehat{m}_{n_{k,1}^{\text{train}}}^l(X_q)\right\}^2\right]}{\sum_{l'=1}^{L} \exp\left[-\kappa \sum_{q=n_{k,1}^{\text{train}}+1}^{i-1} \left\{Y_q - \widehat{m}_{n_{k,1}^{\text{train}}}^{l'}(X_q)\right\}^2\right]} \quad \text{for } n_{k,1}^{\text{train}} + 2 \leq i \leq n_{k,1}, \quad (20)$$

where $\widehat{\Omega}_{l,n_{k,1}^{\text{train}}+1} = 1/L$ and the ensemble predictions for the conditional outcomes under treatment is $\widehat{m}_{1,k}(X_i) = \sum_{l=1}^{L} \widehat{\Omega}_l \widehat{m}_{1,k}^l(X_i)$. The above procedure is then repeated in the control group $D_{k,0}$ and $a = 0$ to obtain the ensemble predictions for the conditional outcomes under control.

The tuning parameter $\kappa$ in (20) can be selected via cross-validation and [4] showed that the performance of model mixture estimators is generally robust across different choices of $\kappa$; they recommended choosing $\kappa = \max(1, \lfloor \log(L) \rfloor)$.

## C Practical considerations

### C.1 Assessing the identification assumptions

In general, our proposed causal identification assumptions in Section 2 are similar to those in existing work in the domain of causal generalizability and transportability [2, 1, 16]. Further, the estimation assumptions that we make for consistency require only one of $J + K$ models to be correctly specified, which is weaker than the other work, requiring one of the outcome and treatment models to be correctly specified [2, 1].

More specifically, some of the assumptions we proposed can be verified empirically. For instance, Assumptions (A3), (A5), and (A7) can be verified by computing the treatment probabilities for specific patient subgroups of interest. Diagnostic plots can be valuable tools for detecting potential positivity violations in practice. For a more comprehensive discussion and practical guidance, one can refer to [1] and [7]. Furthermore, in specific real-world scenarios, these assumptions can also be justified with domain knowledge. For example, in our case study with the CMS dataset, Assumptions (A3) and (A5) hold because each state has some hospitals performing PCI treatment and there are no baseline covariates that a contraindications for the treatment. Assumption (A7) is also plausible since none of the states deny admission to AMI patients on the basis of any of the baseline covariates.

The Assumptions (A2) and (A6) are not testable, so it is reasonable to explore how the conclusion changes with different degrees of assumption violations. Various sensitivity analysis methods can be applied, for example, [13] and [3]. Discerning the plausibility of assumptions can also be facilitated using directed acyclic graphs; the graphical identification algorithms for assessing transportability can be useful [14]. In the important practical scenario that treatment is randomly assigned (e.g., a multi-site randomized trial), then consistency, exchangeability, and positivity of treatment assignment will hold by design.

When identification assumptions do not hold, we cannot identify the TATE from the observed data. However, we can still derive bounds under relaxed identification requirements. These bounds are useful because they provide a range of plausible values for the TATE. In practice, we suggest conducting sensitivity analyses. For example, we can relax identification assumption (A6) as follows:

$$|E\{Y_i(a) \mid V_i = v, R_i = T\} - E\{Y_i(a) \mid V_i = v, R_i = k\}| \leq \delta_k \text{ almost surely for all } a. \quad (21)$$

The specific value for $\delta_k$ can be determined with domain knowledge. Investigators can also learn the value of $\delta_k$ that changes results substantially (e.g., flips the sign of the treatment effect).

## C.2 The target population can be defined generally

In this work, without loss of generality, we assume that our target population is a population of a specific site, namely the target site population. However, our target population can be generally specified to correspond to different goals. For example, a target population can be some covariate profiles of the patients admitted to the hospital or any covariate profiles of patients admitted to hospitals within a geographic region, the definition we delved into during our case study. As one reviewer suggested, we can even define our target population at the level of an individual's covariate profile, with all other individuals being regarded as source individuals. Within this framework, our method can potentially be extended to estimate individual treatment effects. However, we identify some challenges in extending our method to the individual treatment effect estimation. First, obtaining a reliable estimate of the target individual effect to obtain an initial "anchor estimate" is not a simple task given that we perform estimation and inference for only one unit. Another challenge is the privacy concern. If we operate on an individual level, we must be more careful about possible privacy leakage and membership inference attacks.

## C.3 Detecting the model heterogeneity

In practice, it can be challenging to assess whether the same outcome and treatment models should be adopted across sites. Our primary recommendation is to consult with investigators at each site and draw upon their domain-specific knowledge. For example, in the context of hospital quality measurement, a shared model may not be suitable, as certain hospitals might excel in treating specific diagnoses compared to others. Previous research has also indicated that regression models can diverge across sites, even within the same population [10].

In cases where domain-specific expertise is unavailable, we advocate using the model-mixing algorithm by [11] as a useful diagnosis of model heterogeneity across sites. Specifically, we apply the algorithm in each site with a set of candidate outcome and treatment models and examine the associated mixing weight of each model. If substantial variations in mixing weights exist across sites (i.e. certain sites favor one model while others favor a different one), model heterogeneity is likely to occur across sites. In such a case, we recommend the use of the MR-$L_1$ approach over the AIPW-$L_1$ approach.

In the case study, we consider two outcome models and two treatment models. We specify the first model with the patient's demographic variables and the second model with the patient's demographic and comorbidity variables. We extract the model-mixing weights for the two outcome and treatment models and present them in Figure 5. We observe substantial variations in the assigned mixing weights across different states, especially in terms of the treatment model. For instance, in the State of Alabama (AL), the first model receives a mixing weight of 0.24, while the second model is assigned a weight of 0.76. In contrast, in the State of Nebraska (NE), the second model is favored with a mixing weight of 0.84, whereas the first model is assigned a weight of 0.16. These findings suggest that simply assuming the same treatment model across states may not be suitable.

## C.4 Stabilize the density ratio estimation

In practice, some sites can have extreme covariate shifts against the target site, meaning that their patient population can be very different from the target patient population. Under this scenario, correctly specifying the density ratio model can be challenging. To increase the flexibility of the density ratio models, we allow for different basis functions to capture potential nonlinearities. Furthermore, in the implementation, we suggest stabilizing density ratio estimates by trimming extreme values [12].

To further protect against potential density ratio model misspecification, we introduce the adaptive ensemble method, described in Section 4, so that the source site estimates will be down-weighted if they are extremely biased or the variance is too large due to the extreme density ratio weights. Finally, as we have shown in Theorem 2, if we can estimate the $\tau$ function well enough, our site-specific estimator can still be consistent, even when the density ratio models are completely misspecified.

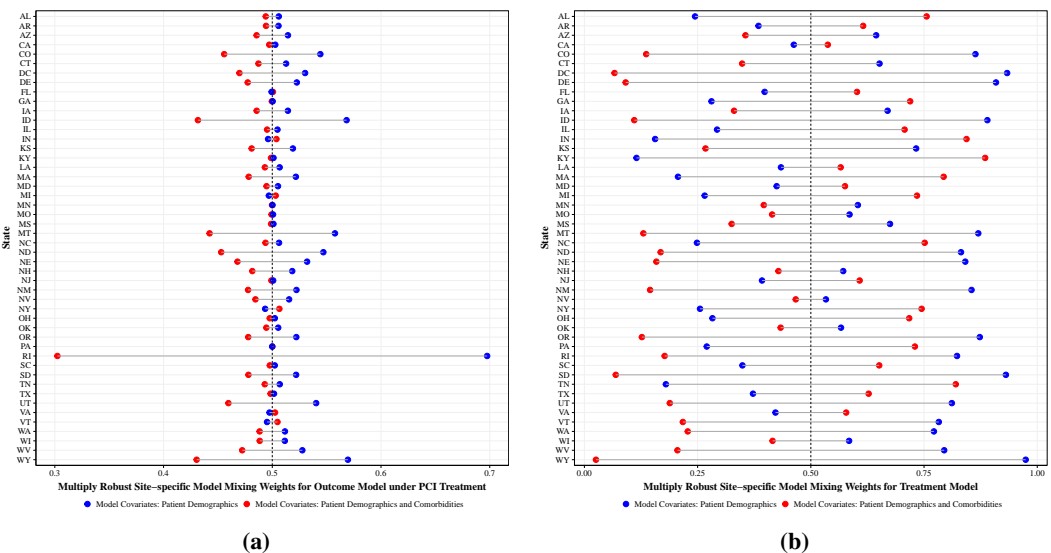

**Figure 5:** Model-mixing weights for the two outcome models and two treatment models across states.

## D    Additional experimental details

We consider five different estimators: (i) an augmented inverse probability weighted (AIPW) estimator using data from the target site only (Target), (ii) an AIPW estimator that weights each site proportionally to its sample size (SS), (iii) an AIPW estimator that weights each site inverse-proportionally to its variance (IVW), (iv) an AIPW estimator that weights each site with the $L_1$ weights defined in (9) (AIPW-$L_1$), and (v) a multiply robust estimator with the $L_1$ weights defined in (9) (MR-$L_1$).

We consider a total of five sites and fix the first site as the target site with a relatively small sample size of 300. The source sites have larger sample sizes of $\{500, 500, 1000, 1000\}$. We model heterogeneity in the covariate distributions across sites with skewed normal distributions and varying levels of skewness in each site, $X_{kp} \sim \mathcal{SN}\left(x; \Xi_{kp}, \Omega_{kp}^2, \gamma_{kp}\right)$, where $k \in \{1, ..., 5\}$ indexes each site and $p \in \{1, ..., 4\}$ indexes the covariates. Specifically, for any given site $k \in \mathcal{K}$, we set the location parameters $\Xi_{kp} = 0$ and the scale parameter $\Omega_{kp} = 1$. Moreover, in the case of the target site, we specifically assign the skewness parameter $\gamma_{kp}$ a value of zero, denoting a symmetrical distribution. However, for source site $k \in \mathcal{S}$, we adopt different values for $\gamma_{kp}$ based on the sample size. If the sample size is equal to 500, we assign $\gamma_{kp}$ the value of $(1/2)^p$, reflecting a positively skewed distribution. On the other hand, if the sample size is 1000, we assign $\gamma_{kp}$ the value of $-(1/2)^p$, indicating a negatively skewed distribution. Following [8], we generate covariates $Z_{kp}$ as non-linear transformation of $X_{kp}$ such that $Z_{k1} = \exp(X_{k1}/2)$, $Z_{k2} = X_{k2}/\{1 + \exp(X_{k1})\} + 10$, $Z_{k3} = (X_{k1}X_{k3}/25 + 0.6)^3$ and $Z_{k4} = (X_{k2} + X_{k4} + 20)^2$.

To obtain the site-specific TATE estimates, we adopt the model-mixing procedure in [11], partitioning the data within each site into two equally sized training and validation datasets five times. The model-mixing weights for outcome models and treatment models are determined by computing 20 and 7, respectively. We follow the recommendation in [11], setting the tuning parameter $\kappa$ in 20 as $\kappa = \max(1, \lfloor \log(L) \rfloor)$.

To compute the optimal ensemble weights, we solve the adaptive LASSO problem 9. The tuning parameter $\lambda$ is selected through cross-validation using a grid of values $\{0, 10^{-3}, 10^{-2}, 0.1, 0.5, 1, 2, 5, 10\}$. To perform cross-validation, the simulated datasets in each

site are split into two equally sized training and validation datasets. To account for cases where source sites have extreme covariate shifts against the target site, we stabilize density ratio estimates by trimming extreme values such that all estimated density ratios are within a range from 0.01 to 100.

## D.1 No covariate mismatch

The main text presented simulation results in Table 1 based on 500 simulations, assuming there is covariate mismatch. Here, we investigate the performance of the estimators assuming no covariate mismatch, i.e. $V_k = X_k = \{X_{k1}, X_{k2}, X_{k3}, X_{k4}\}$ for target and source sites. We generate potential outcomes as

$$Y_k(a) = 210 + X_k\beta_x + Z_k\beta_z + \varepsilon_k \tag{22}$$

where $\beta_x = \beta_z = (27.4, 13.7, 13.7, 13.7)$. For units in the target site, we generate outcomes with $X_k$ only by setting $\beta_z = 0$; for units in the source sites, either $X_k$ or $Z_k$ is used to generate outcomes. If $\beta_x \neq 0$, then $\beta_z = 0$ and vice versa. Similarly, the treatment is generated as

$$A_k \sim \text{Bernoulli}\,(\pi_k) \quad \pi_k = \text{expit}(X_k\alpha_x + Z_k\alpha_z) \tag{23}$$

where $\alpha_x = \alpha_z = (-1, 0.5, -0.25, -0.1)$. For units in the target site, we generate treatments with $X_k$ only by setting $\alpha_z = 0$; for units in the source sites, either $X_k$ or $Z_k$ is used to generate treatments. If $\alpha_x \neq 0$, then $\alpha_z = 0$ and vice versa. With this data generation scheme, the true ATE is $\Delta_T = 0$.

We compare the performance of the five estimators under the following settings ($C$ denotes the proportion of source sites that correctly specify the outcome and treatment models):

**Setting 1 ($C = 0$):** outcomes and treatments in source sites are generated with $Z_k$ by setting $\beta_x = 0$ and $\beta_z = (27.4, 13.7, 13.7, 13.7)$. However, all source sites misspecify both models with $X_k$. The target site correctly specifies both models with $X_k$.

**Setting 2 ($C = 1/2$):** outcomes and treatments are generated with $X_k$ in Sites 2 and Site 4 by setting $\beta_x = (27.4, 13.7, 13.7, 13.7)$ and $\beta_z = 0$. In Sites 3 and 5, outcomes and treatments are generated with $Z_k$ by setting $\beta_x = 0$ and $\beta_z = (27.4, 13.7, 13.7, 13.7)$. Thus, the outcome and treatment models are misspecified in Sites 3 and 5 with $Z_k$, and only half of the source sites correctly specify the models.

**Setting 3 ($C = 1$):** outcomes and treatments in all source sites are generated with $X_k$ by setting $\beta_x = (27.4, 13.7, 13.7, 13.7)$ and $\beta_z = 0$, so all source sites correctly specify outcome and treatment models with $X_k$.

**Table 2:** Mean absolute error (MAE), root mean squared error (RMSE), coverage (Cov.), and length (Len.) of 95% CIs based on 500 simulated data sets in three (mis)specification settings.

|  | Target | SS | IVW | AIPW-$L_1$ | MR-$L_1$ |
|---|---|---|---|---|---|
| $C = 0$ | | | | | |
| MAE | 0.109 | 1.933 | 0.177 | 0.110 | 0.050 |
| RMSE | 0.141 | 1.987 | 0.219 | 0.144 | 0.061 |
| Cov. | 0.950 | 0.998 | 0.826 | 0.936 | 0.960 |
| Len. | 0.551 | 7.035 | 0.567 | 0.547 | 0.234 |
| $C = 1/2$ | | | | | |
| MAE | 0.109 | 1.111 | 0.107 | 0.109 | 0.050 |
| RMSE | 0.141 | 1.189 | 0.139 | 0.140 | 0.062 |
| Cov. | 0.950 | 1.000 | 0.942 | 0.950 | 0.962 |
| Len. | 0.551 | 6.010 | 0.540 | 0.547 | 0.242 |
| $C = 1$ | | | | | |
| MAE | 0.109 | 0.036 | 0.035 | 0.050 | 0.049 |
| RMSE | 0.141 | 0.045 | 0.044 | 0.064 | 0.063 |
| Cov. | 0.950 | 0.968 | 0.956 | 0.958 | 0.960 |
| Len. | 0.551 | 0.195 | 0.191 | 0.260 | 0.253 |

The results in Table 2 indicate that the MR-$L_1$ estimator has lower RMSE than the Target estimator when some source sites have correctly specified models ($C = 1/2$ and $C = 1$). Relative to the MR-$L_1$ estimator, the SS and IVW estimators demonstrate larger biases and RMSE, and lower coverage when some source sites have misspecified models ($C = 0$ and $C = 1/2$). The MR-$L_1$ estimator

shows reduced biases and RMSE compared to the AIPW-$L_1$ estimator, while maintaining similar coverage; this improvement can be attributed to the inclusion of an additional model that closely resembles the true model. When all source sites correctly specify working models ($C = 1$), the IVW estimator performs optimally with the shortest confidence interval as expected.

To provide further insights into the site-specific estimates, we present simulation results in Figure 6. The results show that in cases where some sites fail to specify their working models properly, AIPW estimators exhibit significant bias, while the multiply robust estimators can accurately recover the true TATE. This can be attributed to the fact that the additional candidate model closely approximates the true underlying models. These findings underscore the enhanced safeguard against model misspecification provided by the multiply robust estimators.

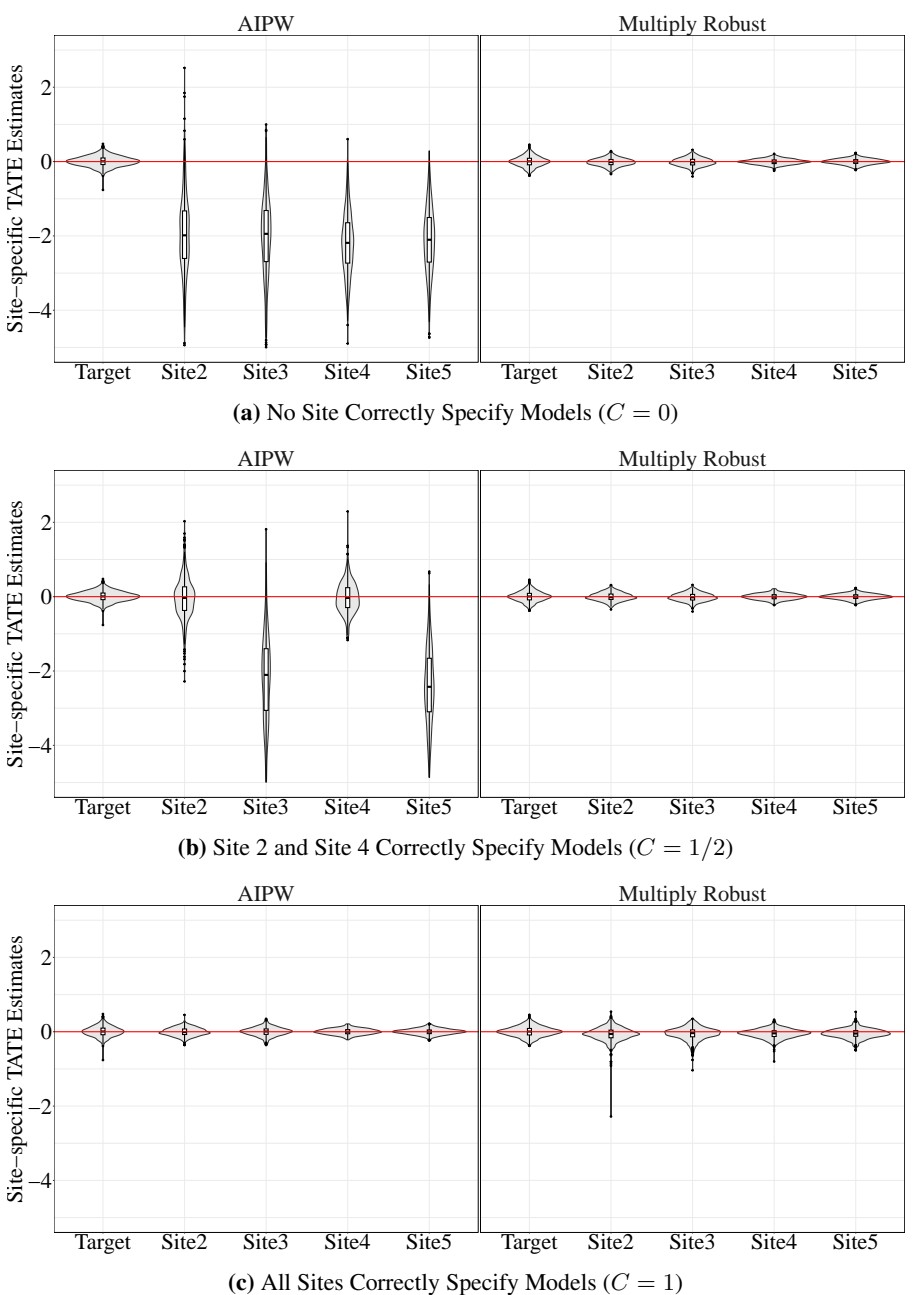

**Figure 6:** Results of site-specific estimation based on 500 simulated data sets in three (mis)specification settings.

## D.2 Non-comparable source site sample sizes

To study if the proposed method is robust in the case where the number of samples in different sites is not comparable, we generated a new experiment building on the one in Section D.1. Specifically, the target site has a sample size of 300, and the two source sites have sample sizes of 300 (Source Site 2) and 3000 (Source Site 3). We conducted 300 iterations of the experiment due to the computing resource constraints. The results in Table 3 indicate that even for non-comparable target and source site sample sizes, the proposed MR-$L_1$ still demonstrates superior performance than other methods.

**Table 3:** Mean absolute error (MAE), root mean squared error (RMSE), coverage (Cov.), and length (Len.) of 95% CIs based on 300 simulated data sets in three specification settings.

|      | Target | SS     | IVW   | AIPW-$L_1$ | MR-$L_1$ |
|------|--------|--------|-------|------------|----------|
| MAE  | 0.111  | 1.938  | 0.177 | 0.110      | 0.070    |
| RMSE | 0.143  | 1.985  | 0.226 | 0.142      | 0.086    |
| Cov. | 0.953  | 1.000  | 0.880 | 0.957      | 0.973    |
| Len. | 0.555  | 15.140 | 0.642 | 0.554      | 0.378    |

**(a)** Setting 1: Only the target site (size of 300) correctly specifies both models

|      | Target | SS     | IVW   | AIPW-$L_1$ | MR-$L_1$ |
|------|--------|--------|-------|------------|----------|
| MAE  | 0.111  | 1.821  | 0.155 | 0.111      | 0.073    |
| RMSE | 0.143  | 1.872  | 0.200 | 0.143      | 0.092    |
| Cov. | 0.953  | 1.000  | 0.907 | 0.953      | 0.947    |
| Len. | 0.555  | 16.100 | 0.627 | 0.554      | 0.387    |

**(b)** Setting 2: Target site (size of 300) and Source Site 2 (size of 300) correctly specify both models

|      | Target | SS    | IVW   | AIPW-$L_1$ | MR-$L_1$ |
|------|--------|-------|-------|------------|----------|
| MAE  | 0.111  | 0.241 | 0.051 | 0.109      | 0.072    |
| RMSE | 0.143  | 0.292 | 0.065 | 0.140      | 0.088    |
| Cov. | 0.953  | 1.000 | 0.990 | 0.953      | 0.973    |
| Len. | 0.555  | 1.716 | 0.374 | 0.545      | 0.382    |

**(c)** Setting 3: Target site (size of 300) and Source Site 3 (size of 3000) correctly specify both models

## D.3 Alternative data generation process

We designed a data generation process based on Section D.1 such that there is extra information in the source sites which is not helpful for specifying models. We conducted two experiments which enabled us to study the efficiency loss due to specifying more models. Specifically, for the first experiment, we generate the source and target site data according to 14 and 15. For units in the target and source sites, we generate outcomes with $X_k$ only by setting $\beta_z = 0$ and $\beta_x = (27.4, 13.7, 13.7, 13.7)$. Similarly, for units in the target and source sites, we generate treatments with $X_k$ only by setting $\alpha_z = 0$ and $\alpha_x = (-1, 0.5, -0.25, -0.1)$. The true ATE is still set to $\Delta_T = 0$. To examine the estimator's performance under covariate mismatch setting, the target site correctly specifies outcome and treatment models with $X_k$, while all source sites specify outcome and treatment models with $X_k$ and $(Z_{k1}, Z_{k2})$; $(Z_{k1}, Z_{k2})$ are two additional dimensions that are not helpful for specifying models. The results in Table 4 indicate that including additional covariates does not lead to a degradation in the performance of the MR-$L_1$ estimator. The MR-$L_1$ estimator demonstrates comparable levels of bias, RMSE, confidence interval coverage, and interval length as observed with the AIPW-$L_1$ estimator. As expected, the MR-$L_1$ estimator outperforms the Target estimator by achieving lower RMSE values, and the IVW estimator attains optimal performance with the shortest confidence interval since all source sites correctly specify the working models.

**Table 4:** Mean absolute error (MAE), root mean squared error (RMSE), coverage (Cov.), and length (Len.) of $95\%$ CIs based on $500$ simulated data sets in covariate mismatch setting.

|        | Target | SS    | IVW   | AIPW-$L_1$ | MR-$L_1$ |
|--------|--------|-------|-------|------------|----------|
| MAE    | 0.109  | 0.036 | 0.035 | 0.050      | 0.050    |
| RMSE   | 0.141  | 0.045 | 0.044 | 0.064      | 0.063    |
| Cov.   | 0.950  | 0.968 | 0.956 | 0.958      | 0.954    |
| Len.   | 0.551  | 0.195 | 0.191 | 0.260      | 0.251    |

### D.4 Reproducibility and computing resources

All experiments in this study were performed using the statistical programming language R (version 4.2.2). The package `sn` (version 2.1.0) was employed for generating covariates that follow a skewed-normal distribution within each site. To solve the density ratio estimating equations, we utilized the `rootSolve` package (version 1.8.2.3). For the estimation of adaptive ensemble weights based on penalized regression of site-specific influence functions, we employed the `glmnet` package (version 4.1-4).

To enhance computational efficiency, parallel computing packages were employed. Specifically, the packages `foreach` (version 1.5.2) and `doParallel` were employed to facilitate the replication of experiments. For the purpose of model-mixing in multiply robust estimation, we employed the `parallel` package (version 4.2.2). The replication of experiments was carried out using ten CPU cores, while the implementation of the model-mixing algorithm utilized five CPU cores.

## E Proofs

### E.1 Proof of Theorem 1

If Assumptions (A1) - (A3) hold, the mean counterfactual outcomes in the target population can be identified using data from the target site;

$$
\begin{aligned}
\mu_{a,T} &= E\left\{Y_i(a) \mid R_i = T\right\} \\
&= E[E\left\{Y_i(a) \mid V_i = v, R_i = T\right\} \mid R_i = T] \\
&= E[E\left\{Y_i(a) \mid V_i = v, A_i = a, R_i = T\right\} \mid R_i = T] \\
&= E[E\left\{Y_i \mid V_i = v, A_i = a, R_i = T\right\} \mid R_i = T].
\end{aligned}
\tag{24}
$$

The second line follows the law of total expectation; the third line follows Assumption (A2); the last line follows Assumption (A1).

If Assumptions (A1), (A4) - (A7) hold, the mean counterfactual outcomes in the target population can be identified using data from source sites;

$$
\begin{aligned}
\mu_{a,T} &= E\left\{Y_i(a) \mid R_i = T\right\} \\
&= E[E\left\{Y_i(a) \mid V_i = v, R_i = T\right\} \mid R_i = T] \\
&= E[E\left\{Y_i(a) \mid V_i = v, R_i = k\right\} \mid R_i = T] \\
&= E\left(E[E\left\{Y_i(a) \mid X_i = x, V_i = v, R_i = k\right\} \mid V_i = v, R_i = k] \mid R_i = T\right) \\
&= E\left(E[E\left\{Y_i(a) \mid X_i = x, R_i = k\right\} \mid V_i = v, R_i = k] \mid R_i = T\right) \\
&= E\left(E[E\left\{Y_i(a) \mid X_i = x, R_i = k, A_i = a\right\} \mid V_i = v, R_i = k] \mid R_i = T\right) \\
&= E\left(E[E\left\{Y_i \mid X_i = x, R_i = k, A_i = a\right\} \mid V_i = v, R_i = k] \mid R_i = T\right).
\end{aligned}
\tag{25}
$$

The second line follows the law of total expectation; the third line follows Assumption (A6); the fourth line follows the law of total expectation; the fifth line follows by our setup that $V_i \subseteq X_i$; the sixth line follows Assumption (A4) the last line follows Assumption (A1).

## E.2  Lemmas in [11]

We restate Lemmas E.1 and E.2 that were proved in [11]. These two lemmas together show that the $L_2$ risks of the multiply robust estimators for $\pi_{a,k}$ and $m_{a,k}$ are bounded by the $L_2$ risks of the model with the smallest risks, plus a negligible remainder term.

**Lemma E.1.** *Suppose that for each $j \in \mathcal{J}$, there exists a constant $0 < \epsilon_j < 1/2$ such that $\epsilon_j < \widehat{\pi}_{a,k}^j(x) < 1 - \epsilon_j$ for all $x$ and some $k \in \mathcal{K}$. Then*

$$E\left(\|\widehat{\pi}_{a,k} - \pi_{a,k}\|^2\right) \le \inf_{j \in \mathcal{J}} \frac{2}{\epsilon_j^2} E\left(\|\widehat{\pi}_{a,k}^j - \pi_{a,k}\|^2\right) + \frac{2\log(J)}{n_k - n_k^{train}} \tag{26}$$

*where $\widehat{\pi}_{a,k}(x) = \sum_{j=1}^J \widehat{\Lambda}_j \widehat{\pi}_{a,k}^j(x)$.*

If the candidate propensity models are parametric and one of them is correctly specified for $\pi_{a,k}$, then $\inf_{j \in \mathcal{J}} \frac{2}{\epsilon_j^2} E(\|\widehat{\pi}_{a,k}^j - \pi_{a,k}\|^2)$ converges at a rate of $1/n$. If the candidate propensity models are non-parametric, then $\inf_{j \in \mathcal{J}} \frac{2}{\epsilon_j^2} E(\|\widehat{\pi}_{a,k}^j - \pi_{a,k}\|^2)$ converges at a rate slower than $1/n$. The remainder term $\frac{2\log(J)}{n_k - n_k^{\text{train}}}$ converges at the rate $1/n_k - n_k^{\text{train}}$, so it vanishes at a faster rate than the statistical risks of candidate models themselves.

**Lemma E.2.** *Suppose we have a continuous outcome and follow the continuous outcome model-mixing algorithm presented in Appendix B. Suppose there exist constants $C_1, C_2 > 0$ such that $\sup_{l \in \mathcal{L}} |\widehat{m}_{a,k}^l(x) - m_{a,k}(x)| \le C_1$ for all $x$ and the subexponential norm of $Y - m_{a,k}(x)$ given $X = x$ is bounded above by $C_2$ for all $x$. Then for $a \in \{0,1\}$*

$$E\left(\|\widehat{m}_{a,k} - m_{a,k}\|^2\right) \le \inf_{l \in \mathcal{L}} E\left(\|\widehat{m}_{a,k}^l - m_{a,k}\|^2\right) + \frac{\log(L)}{\kappa\left(n_{k,a} - n_{k,a}^{train}\right)}$$

*for*

$$0 < \kappa \le \max\left[\frac{1}{16eC_1C_2}, \frac{\exp\left\{C_1\left(8eC_2\right)^{-1}\right\}}{4\mathcal{M}_2\left\{(4eC_2)^{-1}\right\} + 16C_1^2\mathcal{M}_0\left\{(4eC_2)^{-1}\right\}}\right] \tag{27}$$

*where $\mathcal{M}_0(t) = 2\exp\left(2e^2C_2^2t^2\right)$, $\mathcal{M}_2(t) = 16\sqrt{2}C_2^2\exp\left(8e^4C_2^2t^2\right)$ and $e = \exp(1)$.*

## E.3  Influence function of site-specific estimator

In this section, we give the form of the influence functions for the site-specific estimators. The complete derivation is omitted here since it follows very closely to the derivation in [5] and [16]. The primary difference with the derivation in [16] is that we use the density ratio weights $\zeta_k(V)$ instead of the inverse probability of selection weights. Define $Z_i = (Y_i, A_i, X_i, R_i)$ with the (partial) baseline covariates in the target site as $V_i \subseteq X_i$. The general form for our efficient influence function is

$$\xi_{a,k}(Z_i) = \frac{1}{P(R_i = k)}\left[\frac{I(A_i = a, R_i = k)}{\pi_{a,k}(X_i)}\zeta_k(V_i)\{Y_i - m_{a,k}(X_i)\}\right]$$

$$+ \frac{1}{P(R_i = k)}\left[I(R_i = k)\zeta_k(V_i)\{m_{a,k}(X_i) - \tau_{a,k}(V_i)\}\right]$$

$$+ \frac{1}{P(R_i = T)}\left[I(R_i = T)\tau_{a,k}(V_i)\right] - \mu_{a,T}. \tag{28}$$

The target site estimator has the following form,

$$\widehat{\mu}_{a,T} = \frac{1}{n_T}\sum_{i=1}^n\left[\frac{I(A_i = a, R_i = T)}{\pi_{a,T}(V_i)}\{Y_i - m_{a,T}(V_i)\} + I(R_i = T)m_{a,T}(V_i)\right]. \tag{29}$$

This is a standard AIPW estimator whose influence function is derived in [2, 6] as

$$\xi_{a,T}(Z_i) = \frac{1}{P(R_i = T)} \left[ \frac{I(A_i = a, R_i = T)}{\pi_{a,T}(V_i)} \{Y_i - m_{a,T}(V_i)\} + I(R_i = T)m_{a,T}(V_i) \right] - \mu_{a,T}. \tag{30}$$

### E.4 Proof of Theorem 2

We first prove that given the conditions in Theorem 2, the site-specific estimator is a consistent estimator of $\mu_{a,T}$, formalized in the following Lemma:

**Lemma E.3.** *Given Assumptions (A1) - (A7), and E.1 and E.2, if one of (B1) or (B2) and one of (C1) or (C2) hold, then $\widehat{\mu}_{a,k}$ is a consistent estimator for $\mu_{a,T}$.*

*Proof of Lemma E.3.* We divide the proof into four cases and show that $\widehat{\mu}_{a,k}$ achieves consistency.

Case 1: When $\overline{\pi}_{a,k}^j = \pi_{a,k}$ and $\overline{\tau}_{a,k} = \tau_{a,k}$

By assumption, we have $\|\widehat{\pi}_{a,k}^j - \pi_{a,k}\| = o_p(1)$ and $\|\widehat{\tau}_{a,k} - \tau_{a,k}\| = o_p(1)$. Given Lemma E.1 for $\widehat{\pi}_{a,k}$, $\|\widehat{\pi}_{a,k} - \pi_{a,k}\| = o_p(1)$. Define $\overline{m}_{a,k}(x) = \frac{1}{L}\sum_{l=1}^{L} \overline{m}_{a,k}^l(x)$. By definition, $\|\widehat{m}_{a,k}^l - \overline{m}_{a,k}^l\| = o_p(1)$, so $\|\widehat{m}_{a,k} - \overline{m}_{a,k}\| = o_p(1)$. Together with $\|\widehat{\zeta}_k - \overline{\zeta}_k\| = o_p(1)$, we can re-write the site-specific estimator as

$$\widehat{\mu}_{a,k} = \frac{1}{n}\sum_{i=1}^{n} \left[ \frac{n}{n_k} \frac{I(A_i = a, R_i = k)}{\pi_{a,k}(X_i)} \overline{\zeta}_k(V_i)\{Y_i - \overline{m}_{a,k}(X_i)\} \right]$$
$$+ \frac{1}{n}\sum_{i=1}^{n} \left[ \frac{n}{n_k} I(R_i = k)\overline{\zeta}_k(V_i)\{\overline{m}_{a,k}(X_i) - \tau_{a,k}(V_i)\} \right]$$
$$+ \frac{1}{n}\sum_{i=1}^{n} \frac{n}{n_T} I(R_i = T)\tau_{a,k}(V_i)$$
$$+ o_p(1) \tag{31}$$

By assumption of i.i.d. units within each site and the law of large numbers, $\widehat{\mu}_{a,k}$ converges in probability to

$$E\left[ \frac{n}{n_k} \frac{I(A_i = a, R_i = k)}{\pi_{a,k}(X_i)} \overline{\zeta}_k(V_i)\{Y_i - \overline{m}_{a,k}(X_i)\} \right]$$
$$+ E\left[ \frac{n}{n_k} I(R_i = k)\overline{\zeta}_k(V_i)\{\overline{m}_{a,k}(X_i) - \tau_{a,k}(V_i)\} \right]$$
$$+ E\left[ \frac{n}{n_T} I(R_i = T)\tau_{a,k}(V_i) \right]$$
$$= \underbrace{E\left[ \frac{n}{n_k} I(R_i = k)\overline{\zeta}_k(V_i)\left\{ \frac{I(A_i = a)}{\pi_{a,k}(X_i)}Y_i - \tau_{a,k}(V_i) \right\} \right]}_{T_1}$$
$$- \underbrace{E\left[ \frac{n}{n_k} I(R_i = k)\overline{\zeta}_k(V_i)\left\{ \frac{I(A_i = a)}{\pi_{a,k}(X_i)} - 1 \right\}\overline{m}_{a,k}(X_i) \right]}_{T_2}$$
$$+ \underbrace{E\left[ \frac{n}{n_T} I(R_i = T)\tau_{a,k}(V_i) \right]}_{T_3}. \tag{32}$$

Given that $\pi_{a,k}$ is the true propensity score model, $E(T_2) = 0$ since

$$E\left\{ \frac{I(A_i = a)}{\pi_{a,k}(X_i)} - 1 \mid X_i, R_i = k \right\} = \frac{P(A_i = a \mid X_i, R_i = k)}{\pi_{a,k}(X_i)} - 1 = 0. \tag{33}$$

Similarly, given that $\tau_{a,k} = E\{m_{a,k}(X_i) \mid V_i, R_i = k\}$ is the true model,

$$
\begin{aligned}
E(T_3) &= E\{E(T_3 \mid R_i = T)\} \\
&= \frac{n}{n_T} P(R_i = T) E(\tau_{a,k}(V_i) \mid R_i = T) \\
&= E(\tau_{a,k}(V_i) \mid R_i = T) \\
&= \mu_{a,T}.
\end{aligned}
\tag{34}
$$

Finally, we consider $T_1$;

$$
\begin{aligned}
E(T_1) &= E\left[\frac{n}{n_k} I(R_i = k)\overline{\zeta}_k(V_i)\left\{\frac{I(A_i = a)}{\pi_{a,k}(X_i)}Y_i - \tau_{a,k}(V_i)\right\}\right] \\
&= E\left(E\left[\frac{n}{n_k} I(R_i = k)\overline{\zeta}_k(V_i)\left\{\frac{I(A_i = a)}{\pi_{a,k}(X_i)}Y_i - \tau_{a,k}(V_i)\right\} \mid V_i\right]\right) \\
&= E\left(\frac{n}{n_k} I(R_i = k)\overline{\zeta}_k(V_i)\left[E\left\{\frac{I(A_i = a)}{\pi_{a,k}(X_i)}Y_i \mid V_i\right\} - \tau_{a,k}(V_i)\right]\right) \\
&= E\left(\frac{n}{n_k} I(R_i = k)\overline{\zeta}_k(V_i)\left[E\left\{\frac{I(A_i = a)}{\pi_{a,k}(X_i)}Y_i \mid V_i, R_i = k\right\} - \tau_{a,k}(V_i)\right]\right) \\
&= E\left\{\frac{n}{n_k} I(R_i = k)\overline{\zeta}_k(V_i)\left(E\left[E\left\{\frac{I(A_i = a)}{\pi_{a,k}(X_i)}Y_i \mid X_i, R_i = k\right\} \mid V_i, R_i = k\right] - \tau_{a,k}(V_i)\right)\right\} \\
&= E\left(\frac{n}{n_k} I(R_i = k)\overline{\zeta}_k(V_i)\left[E\left\{E(Y_i \mid A_i = a, X_i, R_i = k) \mid V_i, R_i = k\right\} - \tau_{a,k}(V_i)\right]\right) \\
&= E\left(\frac{n}{n_k} I(R_i = k)\overline{\zeta}_k(V_i)\left[E\left\{m_{a,k}(X_i) \mid V_i, R_i = k\right\} - \tau_{a,k}(V_i)\right]\right) \\
&= 0, \ \text{ which completes the proof for Case 1.}
\end{aligned}
\tag{35}
$$

Case 2: When $\overline{\pi}_{a,k}^j = \pi_{a,k}$ and $\overline{\zeta}_k = \zeta_k$

By assumption, we have $\|\widehat{\pi}_{a,k}^j - \pi_{a,k}\| = o_p(1)$ and $\|\widehat{\zeta}_k - \zeta_k\| = o_p(1)$. Given Lemma E.1 for $\widehat{\pi}_{a,k}$, $\|\widehat{\pi}_{a,k} - \pi_{a,k}\| = o_p(1)$. Define $\overline{m}_{a,k}(x) = \frac{1}{L}\sum_{l=1}^{L} \overline{m}_{a,k}^l(x)$. By definition, $\|\widehat{m}_{a,k}^l - \overline{m}_{a,k}^l\| = o_p(1)$, so $\|\widehat{m}_{a,k} - \overline{m}_{a,k}\| = o_p(1)$. Together with $\|\widehat{\tau}_{a,k} - \overline{\tau}_{a,k}\| = o_p(1)$, we can write the site-specific estimator as

$$
\begin{aligned}
\widehat{\mu}_{a,k} = &\frac{1}{n}\sum_{i=1}^{n}\left[\frac{n}{n_k}\frac{I(A_i = a, R_i = k)}{\pi_{a,k}(X_i)}\zeta_k(V_i)\left\{Y_i - \overline{m}_{a,k}(X_i)\right\}\right] \\
&+ \frac{1}{n}\sum_{i=1}^{n}\left[\frac{n}{n_k}I(R_i = k)\zeta_k(V_i)\left\{\overline{m}_{a,k}(X_i) - \overline{\tau}_{a,k}(V_i)\right\}\right] \\
&+ \frac{1}{n}\sum_{i=1}^{n}\frac{n}{n_T}I(R_i = T)\overline{\tau}_{a,k}(V_i) \\
&+ o_p(1)
\end{aligned}
\tag{36}
$$

By assumption of i.i.d. units within each site and the law of large numbers, $\widehat{\mu}_{a,k}$ converges in probability to

$$
\begin{aligned}
&E\left[\frac{n}{n_k}\frac{I(A_i = a, R_i = k)}{\pi_{a,k}(X_i)}\zeta_k(V_i)\left\{Y_i - \overline{m}_{a,k}(X_i)\right\}\right] \\
&+E\left[\frac{n}{n_k}I(R_i = k)\zeta_k(V_i)\left\{\overline{m}_{a,k}(X_i) - \overline{\tau}_{a,k}(V_i)\right\}\right] \\
&+E\left[\frac{n}{n_T}I(R_i = T)\overline{\tau}_{a,k}(V_i)\right]
\end{aligned}
$$

$$= E\left\{\underbrace{\frac{n}{n_k}\frac{I(A_i = a, R_i = k)}{\pi_{a,k}(X_i)}\zeta_k(V_i)Y_i}_{T_1}\right\}$$

$$+ E\left[\underbrace{\frac{n}{n_k}I(R_i = k)\zeta_k(V_i)\overline{m}_{a,k}(X_i)\left\{1 - \frac{I(A_i = a)}{\pi_{a,k}(X_i)}\right\}}_{T_2}\right]$$

$$+ E\left[\underbrace{\overline{\tau}_{a,k}(V_i)\left\{\frac{n}{n_T}I(R_i = T) - \frac{n}{n_k}I(R_i = k)\zeta_k(V_i)\right\}}_{T_3}\right] \tag{37}$$

Given that $\pi_{a,k}$ is the true propensity score model, $E(T_2) = 0$ since

$$E\left\{\frac{I(A_i = a)}{\pi_{a,k}(X_i)} - 1 \mid X_i, R_i = k\right\} = \frac{P(A_i = a \mid X_i, R_i = k)}{\pi_{a,k}(X_i)} - 1 = 0. \tag{38}$$

Also since $\zeta_k = f(V_i|R_i = T)/f(V_i|R_i = k)$ is the true density ratio model,

$$E(T_3) = E\left(E\left[\overline{\tau}_{a,k}(V_i)\left\{\frac{n}{n_T}I(R_i = T) - \frac{n}{n_k}I(R_i = k)\zeta_k(V_i)\right\} \mid V_i\right]\right)$$

$$= E\left(\overline{\tau}_{a,k}(V_i)\left[\frac{n}{n_T}E\left\{I(R_i = T)\right\} - \frac{n}{n_k}E\left\{I(R_i = k)\right\}\zeta_k(V_i) \mid V_i\right]\right)$$

$$= 0 \text{ following the results from Appendix A.} \tag{39}$$

Thus, we are left with $T_1$, which equals to $\mu_{a,T}$ by iterative conditioning and Assumptions (A1), (A2), (A4) and (A6).

Proofs for Case 3, when $\overline{m}^l_{a,k} = m_{a,k}$ and $\overline{\tau}_{a,k} = \tau_{a,k}$ and Case 4, when $\overline{m}^l_{a,k} = m_{a,k}$ and $\overline{\zeta}_k = \zeta_k$, follow closely to the proofs of Cases 1 and 2 and are omitted here for brevity. $\square$

*Proof of Theorem 2.* Given the general form of influence function (28), we can decompose the estimation risk of the site-specific estimator as

$$\mathbb{P}_n(\widehat{\mu}_{a,k}) - E(\mu_{a,T}) = (\mathbb{P}_n - E)(\widehat{\mu}_{a,k} - \mu_{a,T}) + (\mathbb{P}_n - E)\mu_{a,T} + E(\widehat{\mu}_{a,k} - \mu_{a,T}) \tag{40}$$

Assuming proper sample splitting methods are employed, as stated in Lemma 2 of [9], we have:

$$(\mathbb{P}_n - E)(\widehat{\mu}_{a,k} - \mu_{a,T}) = o_p(n^{-1/2}). \tag{41}$$

Then, we consider $(\mathbb{P}_n - E)\mu_{a,T}$. By the Central Limit Theorem, we have $\sqrt{n}\{\mathbb{P}_n(\mu_{a,T}) - E(\mu_{a,T})\} \xrightarrow{d} \mathcal{N}(0, \sigma^2)$, where $\sigma^2$ is given by:

$$\sigma^2 = E\left\{\xi_{a,k}(Z_i)^2\right\}$$

$$= E\left\{\frac{1}{P^2(R_i = k)}\left[\frac{I(A_i = a, R_i = k)}{\pi_{a,k}(X_i)}\zeta_k(V_i)\{Y_i - m_{a,k}(X_i)\}\right]^2\right\}$$

$$+ E\left\{\frac{1}{P^2(R_i = k)}\left[I(R_i = k)\zeta_k(V_i)\{m_{a,k}(X_i) - \tau_{a,k}(V_i)\}\right]^2\right\}$$

$$+ E\left\{\frac{1}{P^2(R_i = T)}\left[I(R_i = T)\{\tau_{a,k}(V_i) - \mu_{a,T}\}\right]^2\right\}$$

$$+ \text{ Remaining cross-terms.} \tag{42}$$

It can be verified that all remaining cross-terms have an expected value of zero. Hence,

$$\sigma^2 = E\left\{\frac{1}{P^2(R_i = k)}\left[\frac{I(A_i = a, R_i = k)}{\pi^2_{a,k}(X_i)}\zeta^2_k(V_i)\{Y_i - m_{a,k}(X_i)\}^2\right]\right\}$$

$$+ E\left\{\frac{1}{P^2(R_i = k)}I(R_i = k)\left[\zeta_k(V_i)\{m_{a,k}(X_i) - \tau_{a,k}(V_i)\}\right]^2\right\}$$

$$+ E\left[\frac{1}{P^2(R_i = T)}I(R_i = T)\{\tau_{a,k}(V_i) - \mu_{a,T}\}^2\right]$$

$$= E\left[E\left\{\frac{1}{P^2(R_i = k)}\left[\frac{I(A_i = a, R_i = k)}{\pi_{a,k}^2(X_i)}\zeta_k^2(V_i)\{Y_i - m_{a,k}(X_i)\}^2\right] \mid X_i\right\}\right]$$

$$+ E\left[E\left\{\frac{1}{P^2(R_i = k)}I(R_i = k)\zeta_k^2(V_i)\{m_{a,k}(X_i) - \tau_{a,k}(V_i)\}^2 \mid V_i\right\}\right]$$

$$+ E\left[\frac{1}{P^2(R_i = T)}I(R_i = T)\{\tau_{a,k}(V_i) - \mu_{a,T}\}^2\right]$$

$$= E\left[\frac{P(R_i = k \mid X_i)}{P^2(R_i = k)}\zeta_k^2(V_i)E\left\{\frac{I(A_i = a)}{\pi_{a,k}^2(X_i)}\{Y_i - m_{a,k}(X_i)\}^2 \mid X_i, A_i = a, R_i = k\right\}\right]$$

$$+ E\left\{\frac{P(R_i = k \mid X_i)}{P^2(R_i = k)}\zeta_k^2(V_i)E\left[\{m_{a,k}(X_i) - \tau_{a,k}(V_i)\}^2 \mid X_i, R_i = k\right]\right\}$$

$$+ E\left(\frac{P(R_i = T)}{P^2(R_i = T)}E\left[\{\tau_{a,k}(V_i) - \mu_{a,T}\}^2 \mid R_i = T\right]\right)$$

$$= E\left[\frac{P(R_i = k \mid V_i)}{P^2(R_i = k)}\zeta_k^2(V_i)\frac{\text{Var}\{Y_i \mid X_i, A_i = a, R_i = k\}}{\pi_{a,k}(X_i)}\right]$$

$$+ E\left[\frac{P(R_i = k \mid V_i)}{P^2(R_i = k)}\zeta_k^2(V_i)\text{Var}\{m_{a,k}(X_i) \mid V_i, R_i = k\}\right]$$

$$+ E\left[\frac{\text{Var}\{\tau_{a,k}(V_i) \mid R_i = T\}}{P(R_i = T)}\right]. \tag{43}$$

This expression represents the semiparametric efficiency bound for estimating $\mu_{a,T}$ and is finite given that all nuisance functions are uniformly bounded. Consequently, we can conclude that $(\mathbb{P}_n - E)(\mu_{a,T}) = O_p(n^{-1/2})$.

Moving forward, we analyze the conditional bias term. Firstly, let us define $E\{\widehat{m}_{a,k}(X_i) \mid V_i, R_i = k\} = \tilde{\tau}_{a,k}(V_i)$. Given conditions in Theorem 2 and Lemma E.3, we can rewrite the conditional bias as follows:

$$E(\widehat{\mu}_{a,k} - \mu_{a,T}) = E\left(\frac{n}{n_k}\left[\frac{I(A_i = a, R_i = k)}{\widehat{\pi}_{a,k}(X_i)}\widehat{\zeta}_k(V_i)\{m_{a,k}(X_i) - \widehat{m}_{a,k}(X_i)\}\right]\right)$$

$$+ E\left(\frac{n}{n_k}\left[I(R_i = k)\widehat{\zeta}_k(V_i)\{\tilde{\tau}_{a,k}(V_i) - \widehat{\tau}_{a,k}(V_i)\}\right]\right)$$

$$+ E\left(\frac{n}{n_T}\left[I(R_i = T)\{\widehat{\tau}_{a,k}(V_i) - \mu_{a,T}(V_i)\}\right]\right)$$

$$= E\left(\frac{n}{n_k}\left[\frac{I(A_i = a, R_i = k)}{\widehat{\pi}_{a,k}(X_i)}\widehat{\zeta}_k(V_i)\{m_{a,k}(X_i) - \widehat{m}_{a,k}(X_i)\}\right]\right)$$

$$+ E\left(\frac{n}{n_k}\left[I(R_i = k)\widehat{\zeta}_k(V_i)\{\tilde{\tau}_{a,k}(V_i) - \tau_{a,k}(V_i)\}\right]\right)$$

$$- E\left(\frac{n}{n_k}\left[I(R_i = k)\widehat{\zeta}_k(V_i)\{\widehat{\tau}_{a,k}(V_i) - \tau_{a,k}(V_i)\}\right]\right)$$

$$+ E\left(\frac{n}{n_T}\left[I(R_i = T)\{\widehat{\tau}_{a,k}(V_i) - \mu_{a,T}(V_i)\}\right]\right)$$

$$= E\left(\frac{n}{n_k}\left[\frac{I(A_i = a, R_i = k)}{\widehat{\pi}_{a,k}(X_i)}\widehat{\zeta}_k(V_i)\{m_{a,k}(X_i) - \widehat{m}_{a,k}(X_i)\}\right]\right)$$

$$+ E\left(\frac{n}{n_k}\left[I(R_i = k)\widehat{\zeta}_k(V_i)\{\widehat{m}_{a,k}(X_i) - m_{a,k}(X_i)\}\right]\right)$$

$$- E\left(\frac{n}{n_k}\left[I(R_i = k)\widehat{\zeta}_k(V_i)\left\{\widehat{\tau}_{a,k}(V_i) - \tau_{a,k}(V_i)\right\}\right]\right)$$

$$+ E\left(\frac{n}{n_T}\left[I(R_i = T)\left\{\widehat{\tau}_{a,k}(V_i) - \mu_{a,T}(V_i)\right\}\right]\right)$$

$$= \underbrace{E\left[\frac{n}{n_k}\widehat{\zeta}_k(V_i)\left\{\frac{I(A_i = a, R_i = k)}{\widehat{\pi}_{a,k}(X_i)} - I(R_i = k)\right\}\left\{m_{a,k}(X_i) - \widehat{m}_{a,k}(X_i)\right\}\right]}_{T_1}$$

$$+ \underbrace{E\left[\left\{\frac{n}{n_T}I(R_i = T) - \frac{n}{n_k}I(R_i = k)\widehat{\zeta}_k(V_i)\right\}\left\{\widehat{\tau}_{a,k}(V_i) - \tau_{a,T}(V_i)\right\}\right]}_{T_2}. \quad (44)$$

Step 3 is derived from Step 2 through conditioning on the variables $(V_i, R_i = k)$, and we obtain the following equality:

$$E\left(\frac{n}{n_k}\left[I(R_i = k)\widehat{\zeta}_k(V_i)\left\{\widetilde{\tau}_{a,k}(V_i) - \tau_{a,k}(V_i)\right\}\right]\right) = E\left(\frac{n}{n_k}\left[I(R_i = k)\widehat{\zeta}_k(V_i)\left\{\widehat{m}_{a,k}(V_i) - m_{a,k}(V_i)\right\}\right]\right). \quad (45)$$

Step 4 can be derived from Step 3 by definition; in the target site, the conditional outcome is represented by $\mu_{a,T} = m_{a,T}(V_i)$ and $\tau_{a,T} = E\left\{m_{a,T}(V_i) \mid V_i, R_i = T\right\} = m_{a,T}(V_i) = \mu_{a,T}$. Conditioning on $X_i$, we have the following expression:

$$T_1 = E\left[\frac{n}{n_k}\widehat{\zeta}_k(V_i)\left\{\frac{I(A_i = a, R_i = k)}{\widehat{\pi}_{a,k}(X_i)} - I(R_i = k)\right\}\left\{m_{a,k}(X_i) - \widehat{m}_{a,k}(X_i)\right\}\right]$$

$$= E\left[\frac{n}{n_k}\widehat{\zeta}_k(V_i)E\left\{\frac{I(A_i = a, R_i = k)}{\widehat{\pi}_{a,k}(X_i)} - I(R_i = k) \mid X_i\right\}\left\{m_{a,k}(X_i) - \widehat{m}_{a,k}(X_i)\right\}\right]$$

$$= E\left[\frac{n}{n_k}\widehat{\zeta}_k(V_i)P(R_i = k \mid X_i)\left\{\frac{\pi_{a,k}(X_i)}{\widehat{\pi}_{a,k}(X_i)} - 1\right\}\left\{m_{a,k}(X_i) - \widehat{m}_{a,k}(X_i)\right\}\right]$$

$$= E\left[\frac{\widehat{\zeta}_k(V_i)}{\widehat{\pi}_{a,k}(X_i)}\left\{\pi_{a,k}(X_i) - \widehat{\pi}_{a,k}(X_i)\right\}\left\{m_{a,k}(X_i) - \widehat{m}_{a,k}(X_i)\right\}\right] \quad (46)$$

Given the boundedness of $\widehat{\zeta}_k$ and $\widehat{\pi}_{a,k}$, it can be concluded that $T_1 = O_p(\|\widehat{\pi}_{a,k} - \pi_{a,k}\|\|\widehat{m}_{a,k} - m_{a,k}\|)$. Now let us consider $T_2$. Conditioning on $V_i$, we have the following expression:

$$T_2 = E\left[\left\{\frac{n}{n_T}I(R_i = T) - \frac{n}{n_k}I(R_i = k)\widehat{\zeta}_k(V_i)\right\}\left\{\widehat{\tau}_{a,k}(V_i) - \tau_{a,T}(V_i)\right\}\right]$$

$$= E\left[E\left\{\frac{n}{n_T}I(R_i = T) - \frac{n}{n_k}I(R_i = k)\widehat{\zeta}_k(V_i) \mid V_i\right\}\left\{\widehat{\tau}_{a,k}(V_i) - \tau_{a,T}(V_i)\right\}\right]$$

$$= E\left[\left\{\frac{n}{n_T}P(R_i = T \mid V_i) - \frac{n}{n_k}P(R_i = k \mid V_i)\widehat{\zeta}_k(V_i)\right\}\left\{\widehat{\tau}_{a,k}(V_i) - \tau_{a,T}(V_i)\right\}\right]$$

$$= E\left[\left\{\frac{1}{P(R_i = T)}P(R_i = T \mid V_i) - \frac{1}{P(R_i = k)}P(R_i = k \mid V_i)\widehat{\zeta}_k(V_i)\right\}\left\{\widehat{\tau}_{a,k}(V_i) - \tau_{a,T}(V_i)\right\}\right]$$

$$= E\left[\frac{P(R_i = k \mid V_i)}{P(R_i = k)}\left\{\zeta_k(V_i) - \widehat{\zeta}_k(V_i)\right\}\left\{\widehat{\tau}_{a,k}(V_i) - \tau_{a,T}(V_i)\right\}\right]. \quad (47)$$

Therefore, $T_2 = O_p(\|\widehat{\zeta}_k - \zeta_k\| \|\widehat{\tau}_{a,k} - \tau_{a,k}\|)$.

Combining all results from the above three steps yields

$$\|\widehat{\mu}_{a,k} - \mu_{a,T}\| = O_p\left(n^{-1/2} + \|\widehat{\pi}_{a,k} - \pi_{a,k}\|\|\widehat{m}_{a,k} - m_{a,k}\| + \|\widehat{\zeta}_k - \zeta_k\|\|\widehat{\tau}_{a,k} - \tau_{a,k}\|\right). \quad (48)$$

Further, if the nuisance estimators satisfy the following convergence rate

$$\|\widehat{m}_{a,k} - m_{a,k}\| \|\widehat{\pi}_{a,k} - \pi_{a,k}\| = o_p(1/\sqrt{n}), \quad \|\widehat{\zeta}_k - \zeta_k\| \|\widehat{\tau}_{a,k} - \tau_{a,k}\| = o_p(1/\sqrt{n}), \quad (49)$$

then the conditional bias, $E\left(\widehat{\mu}_{a,k} - \mu_{a,T}\right) = o_p(1/\sqrt{n})$ since $O_p(o_p(1/\sqrt{n})) = o_p(1/\sqrt{n})$.

Consequently, by Slutsky's theorem, $\sqrt{n}(\widehat{\mu}_{a,k} - \mu_{a,T}) \overset{d}{\to} \mathcal{N}(0, \sigma^2)$. Here, $\sigma^2$ represents the semiparametric efficiency bound, which has been derived and defined in (43). □

## E.5 Proof of Theorem 3

*Proof.* The proof presented herein closely follows the methodology outlined in [5]; we start by establishing the consistency and asymptotic normality of the global estimator, assuming a fixed $\eta_k$. We then invoke Lemma 4 and 5 in [5], which state that the proposed adaptive estimation for $\eta_k$ as shown in (9) allows for (i) the recovery of the optimal $\bar{\eta}_k$ by the estimator $\widehat{\eta}_k$, and (ii) the uncertainty introduced by $\widehat{\eta}_k$ is negligible when estimating $\Delta_T$. Essentially, we require Assumptions (A1)-(A7) and the assumptions stated in Theorem 2 to hold. The additional regularity conditions we require are the variance and covariance of the influence functions to be properly bounded. Note that we can invoke Lemma 4 in [5] to prove the negligible uncertainty of our proposed estimator because both the optimization problem posed in [5] and the optimization problem in our formulation aim to minimize the asymptotic variance of the global federated estimator while controlling for possible estimation bias due to the introduction of the penalty term. Specifically, to show the equivalence between S.9 in [5] and (9) in our formulation, we first see that the penalty terms are equivalent under assumptions in Theorem 3. We then establish the equivalence of the leading term by comparing the asymptotic variance of the global federated estimator derived in (55) and the S.9 in [5].

Lemma E.3 demonstrates the consistency of the site-specific estimators given that the source site $k$ satisfies the conditions outlined in Theorem 2. We denote the set of source sites that fulfill the conditions in Theorem 2 as $\mathcal{S}^*$, and consider a fixed $\eta_k$ such that $\eta_k = 0$ for $k \notin \mathcal{S}^*$, then

$$\widehat{\mu}_{a,G}(\eta_k) = \widehat{\mu}_{a,T} + \sum_{k \in \mathcal{K}} \eta_k\{\widehat{\mu}_{a,k} - \widehat{\mu}_{a,T}\} \tag{50}$$

is consistent for $\mu_{a,T}$ since $\widehat{\mu}_{a,k}$ for $k \in \mathcal{S}^*$ are consistent estimators for $\mu_{a,T}$. Hence, we can establish that $\widehat{\Delta}_G(\eta_k) = \widehat{\mu}_{1,G}(\eta_k) - \widehat{\mu}_{0,G}(\eta_k)$ consistently estimates $\Delta_T = \mu_{1,T} - \mu_{0,T}$. Moving forward, we proceed to examine the asymptotic normality of the global estimator by utilizing the influence functions for the site-specific estimators. First, we rewrite the global estimator with the fixed $\eta_k$ as

$$\widehat{\Delta}_G(\eta_k) = \left(1 - \sum_{k \in \mathcal{S}} \eta_k\right)(\widehat{\mu}_{1,T} - \widehat{\mu}_{0,T}) + \sum_{k \in \mathcal{S}} \eta_k\left(\widehat{\mu}_{1,k} - \widehat{\mu}_{0,k}\right) = \left(1 - \sum_{k \in \mathcal{S}} \eta_k\right)\widehat{\Delta}_T + \sum_{k \in \mathcal{S}} \eta_k\widehat{\Delta}_k \tag{51}$$

In Appendix E.3, the influence functions for the source site and target site estimators have been derived. To facilitate representation, we decompose the influence functions for the source site estimators into two parts, each defined on the target sample and source sample, respectively,

$$\xi_{a,k}^{(1)}(Z_i) = \frac{1}{P(R_i = k)}\left[\frac{I(A_i = a, R_i = k)}{\pi_{a,k}(X_i)}\zeta_k(V_i)\{Y_i - m_{a,k}(X_i)\}\right]$$

$$+ \frac{1}{P(R_i = k)}\left[I(R_i = k)\zeta_k(V_i)\{m_{a,k}(X_i) - \tau_{a,k}(V_i)\}\right]$$

$$\xi_{a,k}^{(2)}(Z_i) = \frac{1}{P(R_i = T)}\left[I(R_i = T)\tau_{a,k}(V_i)\right] - \mu_{a,T}. \tag{52}$$

Here, it should be noted that the expectations of where $\xi_{a,k}^{(1)}(Z_i)$ and $\xi_{a,k}^{(2)}(Z_i)$ are both equal to zero. Furthermore, for the target site estimator, we define:

$$\xi_{a,T}^{(2)}(Z_i) = \frac{1}{P(R_i = T)}\left[\frac{I(A_i = a, R_i = T)}{\pi_{a,T}(V_i)}\{Y_i - m_{a,T}(V_i)\} + I(R_i = T)m_{a,T}(V_i)\right] - \mu_{a,T} \tag{53}$$

and the expectation of $\xi_{a,T}^{(2)}(Z_i)$ is also zero. We denote the influence function for $\widehat{\Delta}_k$ as $\xi_{1,k}^{(1)}(Z_i) - \xi_{0,k}^{(1)}(Z_i) + \xi_{1,k}^{(2)}(Z_i) - \xi_{0,k}^{(2)}(Z_i)$. Similarly, the influence function for $\widehat{\Delta}_T$ is given by $\xi_{1,T}^{(2)}(Z_i) -$

$\xi_{0,T}^{(2)}(Z_i)$. Thus, the influence function for the global estimator evaluated with target and source samples can be expressed as:

$$\widehat{\Delta}_G(\eta_k) - \Delta_T = \left(1 - \sum_{k \in \mathcal{S}} \eta_k\right)\left(\widehat{\Delta}_T - \Delta_T\right) + \sum_{k \in \mathcal{S}} \eta_k \left(\widehat{\Delta}_k - \Delta_T\right)$$

$$= \left(1 - \sum_{k \in \mathcal{S}} \eta_k\right) \frac{1}{n_T} \sum_{i=1}^{N} I(R_i = T) \left\{\xi_{1,T}^{(2)}(Z_i) - \xi_{0,T}^{(2)}(Z_i)\right\}$$

$$+ \left(\sum_{k \in \mathcal{S}} \eta_k\right) \frac{1}{n_T} \sum_{i=1}^{N} I(R_i = T) \left\{\xi_{1,k}^{(2)}(Z_i) - \xi_{0,k}^{(2)}(Z_i)\right\}$$

$$+ \sum_{k \in \mathcal{S}} \eta_k \frac{1}{n_k} \sum_{i=1}^{N} I(R_i = k) \left\{\xi_{1,k}^{(1)}(Z_i) - \xi_{0,k}^{(1)}(Z_i)\right\}$$

$$= \frac{1}{N} \sum_{i=1}^{N} I(R_i = T) \left(1 - \sum_{k \in \mathcal{S}} \eta_k\right) \frac{\left\{\xi_{1,T}^{(2)}(Z_i) - \xi_{0,T}^{(2)}(Z_i)\right\}}{P(R_i = T)}$$

$$+ \frac{1}{N} \sum_{i=1}^{N} I(R_i = T) \left(\sum_{k \in \mathcal{S}} \eta_k\right) \frac{\left\{\xi_{1,k}^{(2)}(Z_i) - \xi_{0,k}^{(2)}(Z_i)\right\}}{P(R_i = T)}$$

$$+ \frac{1}{N} \sum_{k \in \mathcal{S}} \sum_{i=1}^{N} I(R_i = k)\eta_k \frac{\left\{\xi_{1,k}^{(1)}(Z_i) - \xi_{0,k}^{(1)}(Z_i)\right\}}{P(R_i = k)} \tag{54}$$

The asymptotic variance for $\widehat{\Delta}_G(\eta_k)$ equals the variance of the influence function (54). Its derivation is similar to that of 43. Let us denote this asymptotic variance as $\mathcal{V}(\eta_k)$. Under the assumption of i.i.d. units within each site, we have:

$$\mathcal{V}(\eta_k) = \left(1 - \sum_{k \in \mathcal{S}} \eta_k\right)^2 \frac{\text{Var}\left\{\xi_{1,T}^{(2)}(Z_i) - \xi_{0,T}^{(2)}(Z_i) \mid R_i = T\right\}}{P(R_i = T)}$$

$$+ \left(\sum_{k \in \mathcal{S}} \eta_k\right)^2 \frac{\text{Var}\left\{\xi_{1,k}^{(2)}(Z_i) - \xi_{0,k}^{(2)}(Z_i) \mid R_i = T\right\}}{P(R_i = T)}$$

$$+ 2\left(1 - \sum_{k \in \mathcal{S}} \eta_k\right)\left(\sum_{k \in \mathcal{S}} \eta_k\right) \frac{\text{Cov}\left\{\xi_{1,T}^{(2)}(Z_i) - \xi_{0,T}^{(2)}(Z_i), \xi_{1,k}^{(2)}(Z_i) - \xi_{0,k}^{(2)}(Z_i) \mid R_i = T\right\}}{P(R_i = T)}$$

$$+ \sum_{k \in \mathcal{S}} \eta_k^2 \frac{\text{Var}\left\{\xi_{1,k}^{(1)}(Z_i) - \xi_{0,k}^{(1)}(Z_i) \mid R_i = k\right\}}{P(R_i = k)} \tag{55}$$

Under the boundedness conditions of the variance and covariance of the influence functions, this variance is finite. Consequently, we can express the asymptotic distribution of $\sqrt{N}\left(\widehat{\Delta}G(\eta_k) - \Delta T\right)$ as:

$$\sqrt{N}\left(\widehat{\Delta}_G(\eta_k) - \Delta_T\right) \xrightarrow{d} \mathcal{N}\left(0, \mathcal{V}(\eta_k)\right) \tag{56}$$

We further define the optimal adaptive weight $\bar{\eta}_k$ as follows:

$$\bar{\eta}_k = \arg \min_{\eta_k = 0 \forall k \notin \mathcal{S}^*} \mathcal{V}(\eta_k) \tag{57}$$

By leveraging Lemmas 4 and 5 from [5], we can recover the optimal $\bar{\eta}_k$ with negligible uncertainty for estimating $\Delta_T$ if we estimate $\widehat{\eta}_{k,L_1}$ using (9). The consistency of $\widehat{\mathcal{V}}(\widehat{\eta}_{k,L_1})$ follows when we can effectively approximate $\mathcal{V}(\bar{\eta}_k)$ with $\widehat{\mathcal{V}}(\widehat{\eta}_{k,L_1})$. Thus,

$$\sqrt{N/\widehat{\mathcal{V}}(\widehat{\eta}_{k,L_1})}\left(\widehat{\Delta}_G(\widehat{\eta}_{k,L_1}) - \Delta_T\right) \xrightarrow{d} \mathcal{N}(0,1) \tag{58}$$

We now proceed to analyze the efficiency gain resulting from the federation process. The estimator that relies solely on the target data is denoted as $\widehat{\Delta}_T = \widehat{\Delta}_G(\eta_0)$, where $\eta_0$ assigns all weights to the target and none to the source. In contrast, the estimator that leverages the proposed adaptive ensemble approach is denoted as $\widehat{\Delta}_G(\widehat{\eta}_{k,L_1}) = (1 - \sum_{k \in \mathcal{S}} \widehat{\eta}_{k,L_1})\widehat{\Delta}_T + \sum_{k \in \mathcal{S}} \widehat{\eta}_{k,L_1}\widehat{\Delta}_k$. Here, $\widehat{\eta}_{k,L_1}$ can recover the optimal weight $\bar{\eta}_k$ that is associated with the minimum asymptotic variance. Consequently, the variance of $\widehat{\Delta}_G(\widehat{\eta}_{k,L_1})$ is no larger than that of the estimator relying solely on the target data by definition.

To establish that the asymptotic variance of $\widehat{\Delta}_G(\widehat{\eta}_{k,L_1})$ is strictly smaller than that of the estimator based solely on the target data $\widehat{\Delta}_T$, we adapt Proposition 1 in [5] with a modified informative source condition. Specifically, for each source site $s \in \mathcal{S}^*$, we define $\widehat{\Delta}_G(\eta_s)$ a global estimator where $\eta_s$ is the optimal ensemble weight if we only consider target site and this source site $s$. Then the modified informative source condition is $\mathrm{Cov}\left\{\sqrt{N}\widehat{\Delta}_T, \sqrt{N}\left(\widehat{\Delta}_G(\eta_s) - \widehat{\Delta}_T\right)\right\}$, where $\widehat{\Delta}_G(\eta_s) - \widehat{\Delta}_T$ can be expressed as

$$
\widehat{\Delta}_G(\eta_s) - \widehat{\Delta}_T = \widehat{\Delta}_G(\eta_s) - \Delta_T - \left(\widehat{\Delta}_T - \Delta_T\right)
$$
$$
= \frac{1}{N}\sum_{i=1}^{N} I(R_i = T)(1 - \eta_s) \frac{\left\{\xi_{1,T}^{(2)}(Z_i) - \xi_{0,T}^{(2)}(Z_i)\right\}}{P(R_i = T)}
$$
$$
+ \frac{1}{N}\sum_{i=1}^{N} I(R_i = T)\eta_s \frac{\left\{\xi_{1,s}^{(2)}(Z_i) - \xi_{0,s}^{(2)}(Z_i)\right\}}{P(R_i = T)}
$$
$$
+ \frac{1}{N}\sum_{i=1}^{N} I(R_i = s)\eta_s \frac{\left\{\xi_{1,s}^{(1)}(Z_i) - \xi_{0,s}^{(1)}(Z_i)\right\}}{P(R_i = s)}
$$
$$
- \frac{1}{N}\sum_{i=1}^{N} I(R_i = T) \frac{\left\{\xi_{1,T}^{(2)}(Z_i) - \xi_{0,T}^{(2)}(Z_i)\right\}}{P(R_i = T)}
$$
$$
= \frac{1}{N}\sum_{i=1}^{N} I(R_i = T)\eta_s \frac{\left\{\xi_{1,s}^{(2)}(Z_i) - \xi_{0,s}^{(2)}(Z_i) - \xi_{1,T}^{(2)}(Z_i) + \xi_{0,T}^{(2)}(Z_i)\right\}}{P(R_i = T)}
$$
$$
+ \frac{1}{N}\sum_{i=1}^{N} I(R_i = s)\eta_s \frac{\left\{\xi_{1,s}^{(1)}(Z_i) - \xi_{0,s}^{(1)}(Z_i)\right\}}{P(R_i = s)}. \tag{59}
$$

In summary, if there exist a source site $s$ with consistent estimator of $\Delta_T$ and further satisfy $\left|\mathrm{Cov}\left\{\sqrt{N}\widehat{\Delta}_T, \sqrt{N}\left(\widehat{\Delta}_G(\eta_s) - \widehat{\Delta}_T\right)\right\}\right| \geq \varepsilon$ where $\varepsilon$ denotes a positive constant, the variance of $\widehat{\Delta}_G$ is strictly smaller than $\widehat{\Delta}_T$. $\qquad\square$

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
