# OpenReview forum: "Multiply Robust Federated Estimation of Targeted Average Treatment Effects"
_NeurIPS.cc/2023/Conference — NeurIPS 2023 poster_

### Official Review · Reviewer_TqVy · 2023-07-14

**Soundness:** 4 excellent
**Presentation:** 4 excellent
**Contribution:** 3 good
**Rating:** 6
**Confidence:** 4

**Summary:**

In this paper, the authors propose a multiply robust targeted average treatment effect estimator when the data is distributed across multiple sites. The proposed method only requires transmitting statistics among the target site and the source site, without directly sharing their own data.

**Strengths:**

1. The proposed method is multiply robust and can handle covariate mismatches among different sites.



2. The theoretical analysis about the proposed estimator's bias and variance is provided.


3. The experimental results are promising.

**Weaknesses:**

1. My major concern is the privacy issue.

	Transmitting statistics cannot guarantee privacy preservation, especially in the case of estimating targeted average treatment effects, where the source site selects a specific group of individuals and queries this group's statistics in other source sites' datasets. In this case, the targeted site can easily perform a membership inference attack. Usually, it is suggested to adopt differential privacy (DP) [1] method to protect privacy.


	[1] Dwork C, Roth A. The algorithmic foundations of differential privacy[J]. Foundations and Trends® in Theoretical Computer Science, 2014, 9(3–4): 211-407.



2. The experiment setting about covariate mismatch is not convincing.

	In my understanding, covariate mismatch means source sites and target sites have different observed covariates, and actually they share the same causal graph (i.e., the same data generation process). However, as mentioned in line 300, by setting the third and fourth element in $β_x$ and $α_x$ to zero, it actually changes the data generation process of the source site.

	The experimental results under the following setting is suggested: The source and target site data are firstly generated by equations 13 and 14, and then the dimension 3 and 4 of X are deleted in the target site. In other words, in the target site, it has unobserved confounders, i.e., the dimension 3 and 4 of X.

**Questions:**

See weakness 1 and 2

---

> ### Author Rebuttal · Authors · 2023-08-10
>
> * Thank you very much for raising this important issue. The DP perspective is very interesting and we will consider adopting DP methods in future work in order to claim formal privacy guarantees. The current work is privacy-preserving in the sense that the only information that is shared is the following: 1) target site covariate means; 2) source site estimates of the TATE and influence functions. Any given source site is not allowed to specify a specific group and query this group’s statistics in any other source site’s dataset. In this sense, the target site cannot easily perform a membership inference attack. To address your comment, we will discuss these matters in the revised text.
> * Thank you for the comments and suggestions regarding the experiments. In the current work, we consider the case where the true data generating mechanisms are different in the target and source sites. We do so in order to show that our method can handle heterogeneous models across sites. However, the reviewer’s perspective is interesting and we will add the corresponding experiment in the revised manuscript.
> * Following the reviewer's suggestion, we plan to generate the following experiment: the source and target site data are firstly generated by equations 13 and 14, and then we randomly generate another two covariates, $X_5$ and $X_6$ (white noise) in the source sites; we have extra information which is not useful for specifying models correctly. We plan to modify the reviewer's suggested experimental plan so that Assumptions A1-A3 hold for the target site. This is required by our method since we need at least one unbiased TATE estimate to anchor on in the federation step.

---

> > ### Comment · Reviewer_TqVy · 2023-08-15
> >
> > The authors' responses are all the experiment plans, there are no additional experimental results available.

---

> > > ### Author Response · Authors · 2023-08-15
> > >
> > > Thank you for your review and suggestions. Due to space limitations, we were only able to present one page of additional results, which we used to display results from additional experiments suggested by another reviewer. However, we will certainly add the additional experimental setting that you suggested in the revised paper.

---

> > > ### Author Response · Authors · 2023-08-17
> > >
> > > Thank you for the comments and suggestions regarding the experiments.
> > >
> > > Following the reviewer's suggestion, we focus on the setting where we have extra information in the source sites which is not helpful for specifying models correctly. We modify the reviewer's suggested experimental plan so that Assumptions A1-A3 hold for the target site. This is required by our method since we need at least one unbiased TATE estimate to anchor on in the federation step.
> > >
> > > **Experiment 1:**
> > > We slightly modify the suggested experiment building on the one in Section 6.1. More specifically, we generate the source and target site data by equations (13) and (14). For units in the target and source sites, we generate outcomes with $X_k$ only by setting $\beta_z = 0$ and $\beta_x = (27.4, 13.7, 13.7, 13.7)$. Similarly, for units in the target and source sites, we generate treatments with $X_k$ only by setting $\alpha_z = 0$ and $\alpha_x = (−1,  0.5, −0.25,  −0.1)$. The true ATE is $\Delta_{T} = 0$. To examine the estimator's performance under the covariate mismatch setting suggested by the reviewer, the target site correctly specifies outcome and treatment models with $X_k$, while all source sites specify outcome and treatment models with $X_k$ and $(Z_{k1}, Z_{k2})$, two additional dimensions that are not helpful. We conduct 300 iterations of the experiment due to time constraints. The results are as follows:
> > >
> > > |      | Target | SS    | IVW   | AIPW-L1 | MR-L1 |
> > > |:----:|:------:|:-----:|:-----:|:-------:|:-----:|
> > > | MAE  | 0.111  | 0.036 | 0.035 | 0.051   | 0.051 |
> > > | RMSE | 0.143  | 0.045 | 0.044 | 0.065   | 0.064 |
> > > | Cov. | 0.953  | 0.963 | 0.957 | 0.957   | 0.957 |
> > > | Len. | 0.555  | 0.195 | 0.191 | 0.260   | 0.251 |
> > >
> > > **Experiment 2:**
> > > In this experiment, we conduct the suggested experiment building on the one in Section 6.2. More specifically, we generate the source and target site data by equations (13) and (14). For units in the target and source sites, we generate outcomes with $X_k$ only by setting $\beta_z = 0$ and $\beta_x = (27.4, 13.7, 0, 0)$. Similarly, for units in the target and source sites, we generate treatments with $X_k$ only by setting $\alpha_z = 0$ and $\alpha_x = (−1,  0.5, 0, 0)$. Under this setting, the target and source sites have the same data generation process, which is the key difference from the setting in Section 6.2. The true ATE is $\Delta_{T} = 0$. To examine the estimator's performance under the covariate mismatch setting suggested by the reviewer, the target site correctly specifies outcome and treatment models with $(X_{k1}, X_{k2})$, while all source sites specify outcome and treatment models with $(X_{k1}, X_{k2}, X_{k3}, X_{k4})$, two additional dimensions that are not helpful. We conduct 300 iterations of the experiment due to time constraints. The results are as follows:
> > >
> > > |      | Target | SS    | IVW   | AIPW-L1 | MR-L1 |
> > > |:----:|:------:|:-----:|:-----:|:-------:|:-----:|
> > > | MAE  | 0.112  | 0.033 | 0.033 | 0.053   | 0.051 |
> > > | RMSE | 0.139  | 0.042 | 0.041 | 0.067   | 0.064 |
> > > | Cov. | 0.940  | 0.970 | 0.967 | 0.947   | 0.947 |
> > > | Len. | 0.541  | 0.189 | 0.186 | 0.258   | 0.250 |
> > >
> > > **Summary:**
> > > Including additional covariates does not lead to a degradation in the performance of the MR-$L_1$ estimator. In the context of Experiments 1 and 2, the MR-$L_1$ estimator demonstrates comparable levels of bias, RMSE, confidence interval coverage, and interval length as observed with the AIPW-$L_1$ estimator. As expected, the MR-$L_1$ estimator outperforms the Target estimator by achieving lower RMSE values, and the IVW estimator attains optimal performance with the shortest confidence interval since all source sites correctly specify the working models.

---

### Official Review · Reviewer_hesp · 2023-07-24

**Soundness:** 3 good
**Presentation:** 2 fair
**Contribution:** 3 good
**Rating:** 5
**Confidence:** 4

**Summary:**

This paper introduces a new federated approach for conducting multi-source studies, allowing valid average treatment effect inferences to be derived for a target population while preserving privacy and addressing heterogeneity in covariate distributions. The proposed methodology incorporates transfer learning across source and target domains to estimate ensemble weights, enabling efficient combination of information from various source sites. The approach also allows using multiple and different outcome and treatment models at each source.

**Strengths:**

This paper combines ideas from 3 different papers:
1. It uses ideas from "Efficient generalization and transportation" which is on single source + single target to multiple sources + single target. This enables handling covariate mismatch between the sources and the target.
2. It uses ideas from "Demystifying a class of multiply robust estimators" allowing the usage of multiple and different outcome
and treatment models at each source. The source-specific estimator is consistent if any one of the multiple outcome models and any one of the multiple treatment models are correctly specified.
3. It uses the density ratio models and the data-adaptive ensembling ideas from "Federated Adaptive Causal Estimation (FACE) of Target Treatment Effects" and other related works. This results in better precision while preserving individual data privacy by sharing only covariate moments of the target samples.

The novelty of the paper lies in combining these ideas to allow investigators from different source to incorporate site-specific covariate information and domain knowledge and provide increased protection against model misspecification for federated estimation of average treatment effect.

**Weaknesses:**

1. The paper only provides asymptotic guarantees on the proposed estimator. It would help to understand the difficulties in providing finite sample bounds.

2. The paper needs more background and motivation in the introduction. It would be useful to add some real-world motivating examples. Likewise, it would be useful to add simulations using some real-world data. The paper also needs to add a lot more intuition and remarks throughout the main body. It is easy to understand certain choices in the paper if one digs into the three papers mentioned above. However, the paper should be self-sufficient and intuition/explanation throughout would help e.g., after eq (5), eq (6) etc.

3. I have concerns about Assumptions A2 and A4. These assumptions essentially mean no unobserved confounding which is rarely valid in real-life settings. Could ideas from the literature on causal inference that allow unobserved confounding be used here?



**Questions:**

Questions:
1. Can similar ideas be used for individual treatment effect estimation? At a high level, if one were to think of sites as individuals, this would mean using information from other (source) individuals to learn about the target individual.

2. Could the authors give examples when A6 would hold in practice? What if none of the sources satisfy these assumptions? In that case, does the proposed estimator only use the target data?

Suggestions:
1. It would help to summarize how the paper differs from the three papers it builds on. Also, I would encourage restructuring the related work section to make it clear which works are single-site vs multiple-sites as well as where the three main ideas used originally come from.

2. $R_i = T$ on line 87 should be $R_i = 1$ to be consistent with the notation defined previously.

3. Please avoid abbreviations without first stating their full-forms, e.g., AIPW on line 128.

**Limitations:**

No. The authors have neither addresses the limitations as well as the potential negative societal impact of their work. I would encourage the authors to add both of these. Please address the concerns raised above regarding A2 and A4 as well as finite sample guarantees among other limitations.

---

> ### Author Rebuttal · Authors · 2023-08-10
>
> Weaknesses
> 1. When the identification assumptions do not hold, we cannot identify the TATE from the observed data. However, we can still derive bounds under relaxed identification requirements. These bounds are useful because they provide a range of plausible values for the TATE. In practice, we can conduct sensitivity analysis. For example, we can relax assumption (A6):
>
> $| E(Y(a)|V=v, R=T) - E(Y(a)|V=v, R-k) | \leq \delta_k$ almost surely for all a.
>
> The specific value for $\delta_k$ can be determined by consulting domain knowledge experts. When that information is not available, investigators can still learn the value of $\delta_k$ that changes the results substantially (e.g., flips the sign of the treatment effect).
>
> * Discerning the plausibility of assumptions can also be facilitated using directed acyclic graphs. Recent graphical identification algorithms for assessing transportability may be useful (Pearl and Barenboim, 2022).
>
> 2. We will add more background and motivation for our problem in the revised text, as well as real-world motivating examples. We will include an analysis of real data in the revised text.  We will apply our method to study the quality of hospital care provided across multiple US states. Our dataset consists of a representative sample of fee-for-service Medicare beneficiaries who were admitted to short-term acute-care hospitals for AMI. A strength of this dataset is that it is representative of the entire fee-for-service Medicare population in the US.
>
> * We will add more remarks and intuitions throughout the main body of the manuscript. We will provide some intuitions after Eq.4 and Eq.5 in the revised manuscript.
>
> 3. The Reviewer’s concerns about assumptions A2 and A4 are valid, but they are standard assumptions made in causal inference (Zeng et al., 2023; Dahabreh et al., 2020). When the assumptions are violated, we can still derive bounds under relaxed identification requirements. These bounds are useful because they provide a range of plausible values for the TATE, which can still be informative. In the revised text, we will discuss the need to conduct sensitivity analyses of violations of these key assumptions. In the important practical scenario that treatment is randomly assigned (e.g., a multi-site randomized trial), then consistency, exchangeability, and positivity of treatment assignment will hold (marginally or conditionally) by design.
>
> Questions
> 1. Thank you for raising this interesting perspective. Yes, we can define our target of inference to be an individual, defined by his/her covariate profile. In that sense, every other individual could be considered as a source individual, and those individuals could be used to learn about the target individual. A challenge in estimation and inference would be obtaining a reasonable estimate of the target individual to obtain an initial “anchor estimate”. Another challenge to extend this idea for individual treatment effect estimation is the privacy concern. If we operate on an individual level, we must be more careful about possible privacy leakage and membership inference attacks.
> In light of your comment, we will discuss these matters in the revised text.
>
> 2. We choose to assume A6, which is weaker than the common ignorability assumption of {$Y(1),Y(0)$} $\perp A | V, R$. We choose to make this assumption because exchangeability in mean (over treatment and site) is weaker than the assumption of exchangeability in distribution (over treatment and site). As one concrete example, consider a setting where the average treatment effect is 0 in both the target and source site. However, the CATE across racial groups is not the same in the target and source site. Suppose that in the target site, the treatment effect is positive for Asians, negative for African Americans, and null for Caucasians, whereas in the source site, it is null for all racial groups. In this scenario, exchangeability in distribution over race across sites does not hold, but the weaker assumption of mean exchangeability over race across sites does hold. In response to your comment, we will discuss these matters in the revised text.
>
> 2b. The reviewer is correct. If none of the sources satisfy these assumptions, the global estimator will only use the target estimates as the adaptive ensemble weights eta_k will converge to zero if all source estimates are biased.
>
> * Thank you for the very helpful suggestions. We will update the text accordingly.
>
> * Limitations: We will provide more discussion about our limitations. We will elaborate on the fact that high-dimensional covariates are not specifically considered in the current proposal. To accommodate the potentially extreme density ratio weights, we applied strategies such as trimming. We are exploring ways to jointly model the propensity score and density ratio to reduce the dimension of parameters to accommodate high dimensional covariates.
>
> * Potential negative societal impact: Thank you for raising this concern. We apologize for not including the relevant discussion in the manuscript. In our revision, we will note that there is a lack of formal privacy guarantees for the strategies developed in the current literature and in our current work. Future work will aim to provide formal privacy guarantees, in line with best practices and guidance with HIPAA or GDPR, to avoid potential leakage of private information from any individual.
>
> References
> * Pearl, J., & Bareinboim, E. (2022). External validity: From do-calculus to transportability across populations. In Probabilistic and causal inference: The works of Judea Pearl (pp. 451-482).
> * Zeng, Z., Kennedy, E. H., Bodnar, L. M., & Naimi, A. I. (2023). Efficient generalization and transportation. arXiv preprint arXiv:2302.00092.
> * Dahabreh, I. J., Robertson, S. E., Steingrimsson, J. A., Stuart, E. A., & Hernan, M. A. (2020). Extending inferences from a randomized trial to a new target population. Statistics in medicine, 39(14), 1999-2014.

---

> > ### Comment · Reviewer_hesp · 2023-08-15
> >
> > Thank you for your detailed response. I've also gone through the reviews from other reviewers. Based on the response, other reviews, the lack of any evidence for real applications, as well as the acknowledgment of the third weakness I mentioned earlier, I am on the fence about my score. I am lowering my score to 5 for now but will re-assess later.
> >
> > Here are a few comments regarding your response:
> >
> > Weakness:
> > 1. My question was more about finite sample bounds vs asymptotic guarantees and not about the sensitivity analysis. Regarding sensitivity analysis, what would these bounds under relaxed identification requirements look like? It would be useful to know how the resulting sensitivity analysis performs on some real-world data.
> > 2. I concur with Reviewer TqVy: the authors' responses are about plans for adding background, motivation, intuition, and real-world experiments but there is none available to assess as of now. Overall, after revisiting the first section of the paper, I think it may require some major revamp but acceptance.
> > 3. I understand and agree that these assumptions are historically common in the community. However, the recent focus is on moving away from these. Given that, it is important to understand the behavior of the method when these assumptions are violated, i.e., via sensitivity analysis. Alternatively, it is important to see how the proposed method behaves on some real-data where these assumptions may not hold (as also mentioned by reviewer LaCg). Also, while random treatment assignment is an important scenario, so is observational study where random assignment may not hold.

---

> > > ### Author Response · Authors · 2023-08-18
> > >
> > > Thank you for going through our response and our response to the other reviewers. We hope the following addresses your concerns.
> > >
> > > Regarding weakness 1, it is possible to use recently developed methods to bound the AIPW estimator for the ATE under _bounded_ unobserved confounding (Yadlowsky et al., 2022, Armstrong and Kolesár, 2021). Since our proposed MR estimator can be expressed as a linear combination of AIPW estimators, one can use these methods with slight modification (accounting for error in the estimation of combination weights), assuming that unconfoundedness is not too large. Unfortunately, we cannot run the sensitivity analysis at the moment because the authors are physically out of the US and the Centers for Medicare and Medicaid (CMS) data requires access to the data from the US only.
> > >
> > > Regarding weakness 2 and 3, we intend to provide a more comprehensive overview of the dataset. This case study focuses on hospital care quality assessment across US states, utilizing a sample of fee-for-service Medicare beneficiaries admitted for Acute Myocardial Infarction (AMI). We will investigate outcome differences between patients who underwent percutaneous coronary intervention (PCI) and those who didn't, while adjusting for patient case-mix. However, in this multi-state estimation setup, we identify some practical and statistical challenges when comparing treatment quality among states:
> > >
> > > **Privacy-preserving efficient estimation:**
> > > With common methods, cross-state comparisons of state-level AMI treatment quality requires the exchange of individual-level data among states, posing privacy concerns and conflicting with regulations like the Health Insurance Portability and Accountability Act Privacy Rule. With a dataset exceeding 100,000 patient observations, employing iterative estimation methods becomes infeasible. Our proposed method not only prioritizes individual data privacy by sharing only summarized information across states but also optimizes efficiency through a one-round information sharing process.
> > >
> > > **Covariate mismatch across states and hospitals:**
> > > Discrepancies can arise due to variations in privacy regulations across states. States with lenient policies may readily share detailed patient-level data, including historical comorbidity and prescriptions, while more stringent states might only provide basic demographic details like age and gender. Covariate mismatch may also arise from divergent clinical expertise. Within our dataset, North Dakota demonstrates a notably higher mean cardiac tech triad index (averaging 0.7), reflecting greater cardiac technology service availability. In contrast, Georgia exhibits a comparatively lower mean (below 0.2). Consequently, patient information from North Dakota likely encompasses a broader spectrum of cardiovascular metrics compared to that from Georgia.
> > >
> > > **Heterogeneity of patient population:**
> > > In the dataset, we observe that states have different patient profiles. Notably, Massachusetts and New York exhibit considerably older patient populations, averaging around 81 years, in contrast to Nevada and Montana, where the average age hovers around 75.
> > >
> > > **Flexibility of model specification:**
> > > Due to all the factors mentioned above, it would be desirable for the investigators within the state that know best about the local conditions to specify models.
> > >
> > > Regarding the evaluation of identification assumptions, in this CMS dataset, the positivity assumptions of the treatment (A3 and A5) hold because each state has some hospitals perform PCI treatment and there are no baseline covariates that a contraindication for the treatment. The positivity assumption for site selection (A7) is also plausible since none of the states deny admission to AMI patients on the basis of any of the baseline covariates. In the first iteration of data analysis, we empirically justify the positivity assumptions by calculating state-by-state summary statistics, e.g., the proportion of patients treated with PCI across states. We identify none of the proportions close to zero or one; the highest proportion is found in Wyoming, with a value more than 0.45, and the lowest proportion is found in Maine, with a value around 0.25. We further analyze by sub-cohorts defined by baseline covariates and report no evidence of violations of A3 and A5. The exchangeability assumptions will be evaluated with sensitivity analysis. Unfortunately, we cannot run the sensitivity analysis at the moment because the authors are physically out of the US and CMS data requires access to the data from the US only.
> > >
> > > References
> > > - Yadlowsky, S., Namkoong, H., Basu, S., Duchi, J., & Tian, L. (2022). Bounds on the conditional and average treatment effect with unobserved confounding factors. The Annals of Statistics, 50(5), 2587-2615.
> > > - Armstrong, T. B., & Kolesár, M. (2021). Finite‐Sample Optimal Estimation and Inference on Average Treatment Effects Under Unconfoundedness. Econometrica, 89(3), 1141-1177.

---

> > > > ### Comment · Reviewer_hesp · 2023-08-20
> > > >
> > > > I appreciate the authors thoughts on how one could potentially apply sensitivity analysis to their method. I also understand why the authors cannot provide any results on CMS data as of now.
> > > >
> > > > However, my concerns regarding finite sample bounds vs asymptotic guarantees as well as background, motivation, and intuition remain unaddressed.

---

> > > > > ### Author Response · Authors · 2023-08-20
> > > > > **Clarification to the reviewer's remaining concerns**
> > > > >
> > > > > We thank the reviewer for responding to our reply. We apologize for not addressing the outstanding concerns. We attempt to do so now.
> > > > >
> > > > > First, we apologize for the confusion in our earlier response about finite sample bounds. It is indeed possible to provide finite sample bounds on the AIPW estimator for the ATE under unobserved confounding. Recently developed methods by Yadlowsky et al., 2022 and Armstrong and Kolesár, 2021 focus on this problem. Because our proposed multiply robust estimator can be equivalently written as a linear combination of such AIPW estimators, it is possible to use these methods with only minor adjustment, for example, the need to account for uncertainty in the additional nuisance functions.
> > > > >
> > > > > Regarding background and motivation, we will add the following to the paper,
> > > > > "In practical scenarios, the proposed methodology would be particularly valuable in multi-source research settings (e.g., consortia) lacking a central data collection point. These methods would be beneficial for combining dissimilar data sources from various hospitals, whether for patient groups beyond older Medicare patients or for similar endeavors. Additionally, these methods would find utility in other contexts, such as clinics, firms, or schools needing to assess their performance relative to peer institutions. These situations may involve data privacy or propriety concerns that prevent the sharing of detailed, unit-level data."
> > > > >
> > > > > Regarding the reviewer's suggestion to contrast the paper with the related literature it builds upon, we will add the following to the paper,
> > > > > "In contrast to the approach in Han et al. (2021), where they assume a shared outcome regression model co-trained across sites, this paper takes a different stance. Here, we accommodate variations in outcome regression models across sites instead of imposing a uniform model. In the context of hospital quality measurement, a shared outcome model would not be suitable, as certain hospitals might excel in treating specific diagnoses compared to others. Previous research has also indicated that regression models can diverge across sites, even within the same population (Li et al., 2021)."
> > > > >
> > > > > "Compared to Zeng et al. (2023), where they consider a single source site and a single target site, we are able to consider multiple source sites and multiple target sites. To prevent negative transfer, we combine the estimates from multiple sites via a data-adaptive ensembling approach first proposed in Han et al. (2021)."
> > > > >
> > > > > "By deriving an equivalent form of the multiply robust estimator, Li et al. (2020) showed that multiple candidate AIPW models could be combined via mixing weights. We take advantage of this equivalent form to obtain closed-form expressions for the variance of our federated global estimator, which builds upon the theory of semi-parametric efficiency theory and influence functions."
> > > > >
> > > > > Regarding more intuition about the method, we will add the following to the paper,
> > > > > "In transportability studies, preventing negative transfer is critical when there are multiple, potentially biased source sites. If source sites are biased, any variance-weighted estimator (e.g., inverse-variance weighting) would inherit this bias. By examining the MSE of the data-adaptive estimator to the limiting estimand of the target estimator, the MSE can be decomposed into a variance term that can be minimized by a regression of influence functions obtained from the asymptotic linear expansion of the target and source estimates, and an asymptotic bias term. This allows us to rewrite the problem of solving for ensemble weights as an adaptive LASSO problem."
> > > > >
> > > > > New reference
> > > > > Li, S., Cai, T., and Duan, R. (2021). Targeting underrepresented populations in precision medicine: A federated transfer learning approach. arXiv preprint arXiv:2108.12112 .

---

### Official Review · Reviewer_7WXj · 2023-07-25

**Soundness:** 3 good
**Presentation:** 2 fair
**Contribution:** 3 good
**Rating:** 5
**Confidence:** 3

**Summary:**

Authors address estimating a target average treatment effect (TATE) in the setting of multi-site studies that include a target site and several source sites. For the target and each source site, authors propose estimating the TATE; then, TATE estimates are ensembled. Authors allow for source sites to have additional covariates that are not in the treatment site.

Contributions:
- Estimators for the per-site TATE that use covariates in the source sites that are not in the target site
- A procedure for learning density ratios to weight from source to target that do not require sharing individual data
- Asymptotics and CIs for their estimators

Authors also incorporate existing methods to their proposed method:
- Multiply robust estimators (similar to doubly-robust estimators, but where you ensemble different estimators for the usual nuisance parameters)
- Ensembling per-site TATE estimates in a way that reduces estimated variance by using plug-ins of influence functions


**Strengths:**

This work incorporates several ideas to address the setting of estimating the TATE using multiple sites. The proposed method does not require sharing individual-level data, in order to weight from source to target. The method addresses the setting when the source sites have covariates that are not in the treatment site.

**Weaknesses:**

1. Assumptions 2, 4, 6 are similar to but weaker than {Y(1), Y(0)} \perp A | V, R, which is a more common ignorability assumption. Choosing to use Assumptions 2, 4, 6 instead of the more common assumption warrants discussion. For example, in what concrete, practical scenarios would Assumptions 2, 4, 6 hold but not the more common assumption? What is an example of covariates in X but not in V?
2. It’s not clear to me how the covariate mismatch experiments satisfy Assumption 6. That would require E[Y(a) | V, R=k] = E[Y(a) | V] for all sites k in K. But the definition for \beta_z, and thus E[Y(a) | V], is different for the target compared to the source sites, as in the target, \beta_z=0 while in source sites, either \beta_x=0 or \beta_z=0, and it is not obvious to me that their expectations should be the same.
3. The experiments and discussion of experiments could be more thorough; see questions
4. The method for density ratio weighting (3.1) seems like a direct application of covariate balancing. It does not seem particularly novel, but maybe I am missing something.
5. The second equality in Equation (16) in the appendix is only true if there is only one source site.
6. In general, clarity could be improved. The paper could use more explanations for the equations.
For example, in Equation (5), it would be nice to have a discussion of why this specific estimator is proposed. I am guessing it is because it looks like the plug-in of the efficient influence function, but it would be clearer if stated.
7. As another example, in Equation (9), it looks like we are trying to minimize an estimate of the asymptotic MSE. Instead, the paper simply describes the equation as a “penalized regression of site-specific influence functions”, which is not very helpful. It also seems strange to me that each influence function is being applied to every data point, given that the influence function is essentially the effect of a data point on an estimate, but only some data points are used for some estimates. Furthermore, it is also strange to me that the weights in (9) are made to minimize the estimate of the asymptotic MSE, but then the tuning parameter lambda is chosen to minimize cross-validated error. Also, cross-validated error of what? The quantities in this section (influence functions, mu’s) are not directly observable.
8. It is often unclear what the specific contributions are, both in the introduction and in the main text of the paper. For example, in the introduction, “utilizing an adaptive weighting approach that optimally combines estimates from source sites” (lines 51-52) is not descriptive about either the contribution or its novelty. Lines 137-138: “Compared to the transportation estimators in [7, 6], we introduce two additional nuisance functions”, each of which “accounts for covariate shifts across sites” and “covariate mismatch across sites”. From a brief look at the references, there is certainly weighting to adjust for covariate shifts across sites. The specific contribution of various parts of this work is not clear.
9. Notation and writing could be cleaned up: in the experimental setting, A is used for both skewness and treatment. In line 83, V_i \subset X_i gets the point across but does not seem like the right notation. In line 144: "sample samples". HIPAA is mistakenly written as HIPPA (line 21)

**Questions:**

1. Can you provide an example of a realistic setting that satisfies the assumptions?
2. The experiment setting is not clear: Y_k(a) in Equation (13) does not seem to depend on a; is this intentional? When is beta_x=0 vs beta_z=0 for units in the source site (line 275)?
3. The choice of experiment setting is not super clear: e.g. why is there skewness? Which choices in the experiment setting are necessary to make MR-L1 outperform AIPW-L1, or to make L1 ensembling outperform the other ensembling methods?
4. More generally, can you discuss the settings in which specific aspects of the proposed method (using X-V rather than V, the ensembling, etc) should outperform alternatives?
5. In the experiment, when there is cross-validation, what are you measuring to select the best tuning parameter?
6. Why do we need Appendix A?
7. Line 121-122: “we devise a data-adaptive ensembling procedure in Section 4 to screen out sites that significantly violate these assumptions”. But that doesn’t look like what it’s doing; Section 4 looks like it's trying to ensemble estimates to get a low MSE under some assumptions, which is not the same as checking whether sites violate assumptions. Could you explain this connection, or rephrase it?

**Limitations:**

1. Experimental setting is limited: can you include experiments on more realistic settings?
2. It's unclear which parts of the experimental setting are important for demonstrating which parts of the proposed method. The density ratio weighting is not experimentally compared with any other methods.

---

> ### Author Rebuttal · Authors · 2023-08-10
>
> Weaknesses
> 1. As one concrete example, suppose the ATE is 0 in both the target and source. However, the CATE across racial groups is not the same. In the target, the treatment effect is positive for Caucasians but negative for African Americans; in the source, it is null for both. Exchangeability in distribution over race does not hold, but the weaker assumption of mean exchangeability does hold. Regarding examples of covariates in X but not in V, suppose that we have a target with observations collected from an RCT, with $V =$ {$X_1, X_2, X_3$} representing age, gender, and BMI. In the source, data is obtained from EHR and include {$X_1, X_2, X_3$}, as well as an additional ICD-10 code. Here, the ICD-10 code would be in X but not in V.
> 2. We generate four covariates {$X_1, X_2, X_3, X_4$} across sites, where $V =$ {$X_1, X_2$} are shifted by setting their means to zero and skewness non-zero. $X_3$ and $X_4$ are distributed with both mean and skewness zero.
> 3. See below.
> 4. Our approach shares similarities with calibration weighting. However, by using the exponential tilt model, we employ semiparametric efficiency theory to enable a closed-form variance rather than relying on bootstrap.
> 5. The procedure is easily applied in parallel across all sources, so it holds for multiple sources, not just one.
> 6. The estimator is based on the plug-in estimator and one-step correction using influence functions. We will provide a detailed derivation of the estimator in the Appendix.
> 7. If source estimates are biased, a variance-weighted estimator would inherit this bias. By examining the MSE of the data-adaptive estimator to the limiting estimand of the target estimator, the MSE can be decomposed into a variance term that can be minimized by regression of influence functions obtained from the asymptotic linear expansion of the target and source estimates, and an asymptotic bias term. This results in an adaptive LASSO problem. Please refer to the details of the CV in Q5.
> 8. We will clarify our contributions more clearly in the revised text.
> 9. We will modify the notations to be consistent and fix typos.
>
> Questions
> 1. When treatment is randomly assigned, consistency, exchangeability, and positivity will hold by design. When identification assumptions do not hold, we cannot identify the TATE from the observed data. However, we can still derive bounds under relaxed requirements. These bounds are useful because they provide a range of plausible values for the TATE. We can conduct sensitivity analysis. For example, we can relax assumption (A6):
> $| E(Y(a)|V=v, R=T) - E(Y(a)|V=v, R-k) | \leq \delta_k$ almost surely for all a.
> The specific value for $\delta_k$ can be determined by consulting domain knowledge experts.
>
> 2. The outcome model can be written as
> $Y_k(a) = 210 + \Delta A + X_k \beta_x  + Z_k \beta_z  + \epsilon_k$.
> We follow the data generation of Kang & Schafer (2007), where $\Delta$ is fixed at 0, so $\Delta A$ is omitted in Eq.13. When only the target site correctly specifies the working models (C = 0), we generate the outcomes in all source sites with $\beta_x = (0, 0, 0, 0)$ and $\beta_z = (27.4, 13.7, 13.7, 13.7)$.
>
> 3. Skewness allows the distribution of covariates across sites to be heterogeneous. AIPW-L1 only allows homogeneous models, while our proposed MR-L1 allows different specifications. When the true data-generating models are different, MR-L1 will outperform AIPW-L1. SS and IVW are expected to perform worse than L1 methods if a source site with a large sample size is biased.
>
> 4. We will elaborate on the interpretation of results in Section 6.
>
> 5. CV is used to select the tuning parameter in Eq.9. In each site, the data is first split into training and validation data. In the training data, we apply methods detailed in Section 3. For a grid of $\lambda$ = (0, 1e-3, 1e-2, 0.1, 0.5, 1, 2, 5, 10), we solve Eq.9, which gives $\eta^*$. With validation data, we calculate the summary statistics for Eq.9 and plug in parameters estimated from corresponding training data. $\lambda_{\text{opt}}$ corresponds to the $\eta$ that minimizes $Q(\eta)$ in the validation data.
>
> 6. The inverse probability of selection weighting method is widely used, but it does not protect individual data privacy. By showing that the two methods in Appendix A are equivalent, we justify the use of density ratio weighting while not sharing individual data.
>
> 7. Mean exchangeability assumptions are not testable from observed data. In practice, if some source sites violate A1, A4, A5, or A6, the corresponding source estimates are likely to exhibit large bias and variance. Our proposed data-adaptive method would downweight these estimates to avoid negative transfer.
>
> Limitations
> 1. We will include an analysis of real data to study the quality of hospital care across multiple US states, as measured by length of stay for acute myocardial infarction (AMI) patients. Our dataset consists of a representative sample of fee-for-service Medicare beneficiaries who were admitted to short-term acute-care hospitals for AMI.
>
> 2. In the implementation, we employed the density ratio weighting method to achieve covariate balance. It's important to note that we have not utilized inverse variance weighting since it does not protect individual data privacy. In response to your suggestion for comparison with other calibration weighting approaches, we acknowledge their relevance. However, the properties of balancing weights obtained by solving an optimization problem are often hard to analyze; the most common method is to leverage some re-sampling method like bootstrap.
>
> References
> * Pearl, J., & Bareinboim, E. (2022). External validity: From do-calculus to transportability across populations. In Probabilistic and causal inference: The works of Judea Pearl (pp. 451-482).
> * Kang, J. D., & Schafer, J. L. (2007). Demystifying double robustness: A comparison of alternative strategies for estimating a population mean from incomplete data.

---

> > ### Comment · Reviewer_7WXj · 2023-08-14
> >
> > Thank you for your response! It addressed most of my concerns and helped my understanding. I have a few additional questions and comments (numbered as before):
> >
> > Weaknesses:
> > 1. In the concrete medical example, what is a realistic scenario in which age, gender, BMI, and ICD-10 are relevant covariates for mean exchangeability over treatment assignment in source populations (A4), but ICD-10 is not necessary for mean exchangeability over treatment assignment in the target population (A2) and mean exchangeability over site selection (A6)? While treatment can depend on anything so that (A2) and (A4) are fine in isolation, if the sources are simply different hospitals, why is it reasonable to exclude ICD-10 for (A6) in this setting?
> > 6. To clarify, my suggestion is not a bigger proof in the Appendix, but to include a high-level description in the text. Otherwise, it requires a bit more reading between the lines.

---

> > > ### Author Response · Authors · 2023-08-14
> > >
> > > Thank you for your response and for the opportunity to clarify your remaining questions and comments.
> > > 1. In the medical example, it would be reasonable to exclude the ICD-10 code (say, for heart failure) if it is a noisy, error-prone covariate in the target population but not in the source populations. In the source populations, it could be the case that the outcome has been adjudicated by a clinical team so that the ICD-10 code is a good proxy of the true outcome of heart failure. In the target population, since adjudication is costly, it may not be cost-feasible to do adjudication. In this case, one would not want to include the noisy, error-prone ICD-10 code.
> > > 2. Thank you for the clarification. We will include a clearer description in the text to help the reader better comprehend, rather than a more detailed proof in the Appendix.

---

> > > > ### Comment · Reviewer_7WXj · 2023-08-14
> > > >
> > > > Thanks! To be clear, detailed proofs in the Appendix are helpful too, e.g. for my initial comment for weakness 7 ("It also seems strange to me that each influence function is being applied to every data point, given that the influence function is essentially the effect of a data point on an estimate, but only some data points are used for some estimates.").
> > > >
> > > > I am curious what it means for ICD-10 to be modeled as the "same" covariate in the source and target, if it is much noisier in the target but not the source.

---

> > > > > ### Author Response · Authors · 2023-08-14
> > > > >
> > > > > We'll certainly keep the detailed proofs in the Appendix but add more high-level description in the main text to help the reader out.
> > > > >
> > > > > Regarding the ICD-10 question, this could boil down to the quality of the data management team and downstream adjudication team in the two hospitals. The ICD-10 code is meant to capture the same diagnosis in both sites, e.g., heart-failure or not at a certain time point. However, one site may do a better job at accurately documenting the time and decision of the correct code.

---

### Official Review · Reviewer_27aW · 2023-07-26

**Soundness:** 3 good
**Presentation:** 4 excellent
**Contribution:** 3 good
**Rating:** 7
**Confidence:** 3

**Summary:**

This paper focuses on federated ATE estimation. The main challenges lie in deriving valid causal inferences from decentralized data with heterogeneity, different data structures, and privacy constraints. To address these issues, the authors propose a privacy-preserving estimator that allows incorporating site-specific covariate information and adopts an adaptive ensembling approach. This method is more flexible and provides increased protection against model misspecification compared to existing approaches.

**Strengths:**

1) The paper addresses a crucial problem in the field of causal ATE estimation, particularly in the context of multi-site studies and privacy-preserving requirement. Compared with the previous works, the setting is practical. For example, it does not assume that a common set of confounders is observed in all sites.

2) The proposed method is technically sound. The propsoed density ratio models and multiply robust estimator in federated setting are novel.

3) This paper provide theoretical guarantee, including identifiability (Theorem 1), local eatimation consistency (Theorem 2), and global eatimation consistency (Theorem 3).

4) The experimental setting are extensive, and the results basically can support what the authors claim.


**Weaknesses:**

1) The positivity assumption (A3, A5, A7) is necessary for the adopted inverse probability weighting method. However, in federated setting, there may be serious client drift and the positivity assumption may not be satisfied on some clients (sites).

2) It seems that the eatimation consistency requires that one of the multiple outcome or treatment models is correctly specified. I am not sure whether this assumption is too strong in practical.


**Questions:**

I wonder whether the proposed method can still work well if the number of samples in different sites are NOT comparable.

**Limitations:**

My concern centers on the assumptions made in this paper. Please refer to the weakness for details.

---

> ### Author Rebuttal · Authors · 2023-08-10
>
> * Thank you for raising this question. The Reviewer is correct in noting that some sites can have extreme covariate shifts against the target site in practice. Under this scenario, correctly specifying the density ratio model can be challenging. To increase the flexibility of our density ratio models, we allow for different basis functions to capture potential nonlinearities. Furthermore, in our implementation, we stabilize density ratio estimates by trimming extreme values (Liu et al, 2017). To further protect against potential density ratio model misspecification, we also introduce the adaptive ensemble method, described in Section 4, so that the source site estimates will be downweighted if they are extremely biased or the variance is too large due to the extreme density ratio weights. Finally, as we have shown in Theorem 1, if we can estimate the $\tau$ function well enough, our site-specific estimator can still be consistent, even when the density ratio models are completely misspecified. As suggested, we will clarify this in the revised text.
>
> * Thank you for this comment. We wish to provide some clarifications and remarks on the assumptions:
>
> 1. The causal identification assumptions that we invoke are common in the literature; in fact, they are similar or weaker than those in existing work in the domain of causal generalizability and transportability (e.g., Dahabreh et al., 2020; Zeng et al., 2023). We also wish to underscore that the estimation assumptions that we make for consistency require only one of $J+K$ models to be correctly specified, which is weaker than requiring one of two models to be correctly specified in the standard doubly robust scenario.
>
> 2. When the identification assumptions do not hold, we cannot identify the TATE from the observed data. However, we can still derive bounds under relaxed identification requirements. These bounds are useful because they provide a range of plausible values for the TATE, which can still be informative. Also, in practice we can always conduct sensitivity analysis to violations to these key assumptions. For example, we can relax assumption (A6) to the following:
>
> $| E(Y(a)|V=v, R=T) - E(Y(a)|V=v, R-k) | \leq \delta_k$ almost surely for all a.
>
> The specific value for $\delta_k$ can be determined by consulting domain knowledge experts. When that information is not available, investigators can still learn the value of $\delta_k$ that changes the results substantially (e.g., flips the sign of the treatment effect).
>
> 3. For other identification assumptions that are not easily verifiable, the investigator can rely on domain knowledge experts to seek guidance for assessing the plausibility of the assumptions, as discussed above. Discerning the plausibility of these assumptions can also be facilitated using directed acyclic graphs, of which recent graphical identification algorithms for assessing generalizability/transportability may be useful (Pearl and Barenboim, 2022).
>
>
> * Thank you also for this comment. In attention to it, we run a new experiment building on the one in Section 6.1, focusing on the setting with non-comparable target site and source site sample sizes, where the target site has a sample size of 300, and the two source sites have sample sizes of 300 (Source Site 2) and 3000 (Source Site 3).
>
> * We conduct 300 iterations of the experiment due to time constraints. In the revised manuscript, we will generate more iterations and present the results. We find that even for non-comparable target and source site sample sizes, the proposed MR-L1 still demonstrates superior performance than other methods (see Table 1 in the attached PDF).
>
> References
> * Liu, S., Takeda, A., Suzuki, T., & Fukumizu, K. (2017). Trimmed density ratio estimation. Advances in neural information processing systems, 30.
> * Dahabreh, I. J., Robertson, S. E., Steingrimsson, J. A., Stuart, E. A., & Hernan, M. A. (2020). Extending inferences from a randomized trial to a new target population. Statistics in medicine, 39(14), 1999-2014.
> * Zeng, Z., Kennedy, E. H., Bodnar, L. M., & Naimi, A. I. (2023). Efficient generalization and transportation. arXiv preprint arXiv:2302.00092.
> * Pearl, J., & Bareinboim, E. (2022). External validity: From do-calculus to transportability across populations. In Probabilistic and causal inference: The works of Judea Pearl (pp. 451-482).

---

> > ### Comment · Reviewer_27aW · 2023-08-10
> > **Concerns Addressed**
> >
> > Thanks for the response!
> >
> > The added experiments and the clarifies addressed my concerns. I agree with the authors that the assumptions are common in community. Therefore, I keep my positive score and tend to accept this paper.

---

> > > ### Author Response · Authors · 2023-08-14
> > > **Thank you!**
> > >
> > > Thank you for your time and insightful comments! This discussion will be very helpful to improve our revised manuscript.

---

### Official Review · Reviewer_LaCg · 2023-07-27

**Soundness:** 3 good
**Presentation:** 2 fair
**Contribution:** 2 fair
**Rating:** 4
**Confidence:** 4

**Summary:**

This paper considers federated learning for causal inferences using multi-site data. The authors propose an estimator which is multiply-robust (to model misspecification and covariate shift) and privacy-preserving with additional nuisance function estimation. The authors then provide theoretical guarantees for the estimator and show it is at least as efficient as the estimator without federated learning under conditions.


**Strengths:**

This paper aims at the problem of federeated learning for causal inference with multi-site data through a new estimator given by (5) and (8). This estimator is intuitive and has some properties like multiply-robust, and the idea is cool. This method allows covariate shift and covariate mismatch across the sites, which relaxes the classical assumption in causal inference with multiple source data. The flow is clear.

**Weaknesses:**

This paper is mainly built on existing works like references [12,19] and most theoretical results are not surprising. It seems that the main contribution is the use of two additional nuisance functions and the density ratio in (5) when there is covariate mismatch or covariate shift. However, since assumption (A6) requires mean exchangeability over site selection, I'm not sure when will covariate mismatch happens if (A6) holds. That is, if the outcome model is the same across different data sources, what is the scenario that we have different covariates in different data sources in real application? If there is no covariate mismatch, then the estimator given in (5) then reduces to a naive AIPW estimator with importance sampling. The authors do not provide enough motivation for this and there is no real data in numerical experiments.

As for the results of theorem 2, I'm worried about the last term in (11). While the classical double robustness condition is $\Vert\hat{\pi}-\pi\Vert \Vert\hat{m}-m\Vert=o_p(n^{-1/2})$, the last term is the product of estimation error from the density ratio and the expectation of outcome conditional on v. The exponential tilt model for density ratio estimation can easily suffer from model misspecification, and the estimation for the density ratio is known to be vulnerable (large variance). In addition, $\pi$ is a conditional mean whose estimator has no guarantee for fast convergence rate without stronger assumptions, and the discussions about its estimation provided in section 3.3 is quite simple. Thus, my feeling is that this product may generally not be $o_p(n^{-1/2})$, but I understand it's hard to provide further theoretical analysis for this part.

The numerical experiments are hard to follow at first glance. This is partially because the use of $X$ and $Z$ (I thought $Z$ was $V$ at first). It seems $X$ in numerical experiments is not the same as $X$ in the theory part, so the authors may want to modify their notations to make it consistent (for example, by pointing out $X$ and $V$ explicitly in the experiments).

Further, I do not see an easy path for the proposed method and theory to be making an impact in practice. I enjoy the theory. However, the assumptions are somewhat too strong. The authors did not provide evidence for real applications where the assumptions are verifiable.

**Questions:**

There are some questions listed in "Weaknesses", and here are some other questions and typos:

1. the definitions of $\hat{m}$ in (4) and (5) are inconsistent: in (4) it is an expectation conditional on V, while in (5) it is conditioned on X.
2. Line 213: $\bar{\mu}$ not defined
3. In section 5.1, should specify the range of $k$, should it contain $k=T$?
4. Theorem 3 mentions that some regularity conditions needed are provided in appendix, but I cannot easily find them (if exists). The authors should explicitly list them in the appendix before the proof of theorem 3.
5. In the proof of theorem 3, the authors invoke lemma 4 of [12] to show the uncertainty introduced by $\hat{\eta}$ is negligible, but the definition of $\hat{\eta}$ in [12] is different from this paper, since the optimization problem is different, why can lemma 4 still work for this paper?
6. Line 280: C not defined.

---

> ### Author Rebuttal · Authors · 2023-08-10
>
> * We wish to clarify two aspects that are fundamental to the contribution of our paper:
> 1. Covariate mismatch can occur regardless of whether assumption (A6) is met.
> 2. We do not require the outcome models to be the same across different sites.
>
> Based on the availability of covariates and investigators’ modeling priorities, different sites may specify different outcome models. Assumption (A6) requires that all effect modifiers that are distributed differently between sites be measured (Zeng et al., 2023). By conditioning on this set of covariates, (A6) is satisfied.
>
> To provide a real life example, consider the following common scenario where we have a target site with limited observations collected from a randomized control trial, with $V =$ {$X_1, X_2, X_3$} representing age, gender, and, say, BMI. In the source site, data is obtained from electronic health records (EHR) and include {$X_1, X_2$},  as well as additional noisy ICD-10 codes used for billing purposes. In this scenario, we have covariate mismatch, but if the true outcome model only involves {$X_1, X_2$}, (A6) is satisfied.
>
> * We will include an analysis of real data in the revised text. We will study the quality of hospital care provided across multiple US states. Our dataset consists of a representative sample of fee-for-service Medicare beneficiaries who were admitted to short-term acute-care hospitals for AMI. A strength of this dataset is that it is representative of the entire fee-for-service Medicare population in the US.
>
> * The Reviewer is correct in noting that some sites can have extreme covariate shifts against the target site in practice. In our implementation, we stabilize density ratio estimates by trimming extreme values (Liu et al, 2017). We also introduce the adaptive ensemble method, described in Section 4, so that the source site estimates will be downweighted if they are extremely biased or the variance is too large due to extreme weights. Finally, if we can estimate the $\tau$ function well enough, our site-specific estimator is consistent, even when the density ratio models are completely misspecified (Theorem 1).
>
> * We are grateful for your positive assessment of the theory. In regard to assumptions, we wish to provide some remarks:
> 1. The causal identification assumptions that we invoke are similar to those in existing work in the domain of causal generalizability and transportability (e.g., Dahabreh et al., 2020; Zeng et al., 2023). Further, the estimation assumptions that we make for consistency require only one of $J+K$ models to be correctly specified, which is weaker than requiring one of two models to be correctly specified.
>
> 2. When identification assumptions do not hold, we cannot identify the TATE from the observed data. However, we can still derive bounds under relaxed identification requirements. These bounds are useful because they provide a range of plausible values for the TATE. In practice we can conduct sensitivity analysis. For example, we can relax (A6) to:
>
> $| E(Y(a)|V=v, R=T) - E(Y(a)|V=v, R-k) | \leq \delta_k$ almost surely for all a.
>
> The specific value for $\delta_k$ can be determined by consulting domain knowledge experts. Investigators can also learn the value of $\delta_k$ that changes results substantially (e.g., flips the sign of the treatment effect).
>
> 3. Some of the assumptions we proposed can be verified empirically such as the positivity assumptions A3, A5 and A7 through diagnostic plots.
>
> 4. Discerning the plausibility of assumptions can also be facilitated using directed acyclic graphs; graphical identification algorithms for assessing transportability may be useful (Pearl and Barenboim, 2022).
>
> 5. In the important practical scenario that treatment is randomly assigned (e.g., a multi-site randomized trial), then consistency, exchangeability, and positivity of treatment assignment will hold by design.
>
> * Q1: The conditional outcome models $\hat{m}$ are subscripted by $T$ and $k \in S$ to explicitly indicate that they can represent different outcome models in the target site $T$ and in the source site $k \in S$.
>
> * Q2: In Line 191, $\hat{\mu}_G$ is defined. In Line 213, $\bar{\mu}_G$ is defined as the limiting value of $\hat{\mu}_G$.
>
> * Q3: Yes, the Reviewer is correct. It should contain $k = T$ and has been corrected.
>
> * Q4: Thank you for pointing this out; we will correct that in the revised text.
>
> * Q5: Both optimization problems aim to minimize the asymptotic variance of the global federated estimator (first square term in the optimization problem) while controlling for possible estimation bias (penalty term). We can show that S.9 in [12] and Eq.9 in our paper are equivalent optimization problems. First, it can be seen that the penalty terms are equivalent under assumptions in Theorem 2. We can see the equivalence of the leading term by comparing the asymptotic variance of the global federated estimator derived in Supplementary Eq. 51, and S.9 in [12]. We will provide the corresponding derivation in the revised Appendix.
>
> * Q6: Thank you for pointing this out. C is the proportion of source sites with correctly specified working models. We will define C in the revised manuscript.
>
> References
> * Zeng, Z., Kennedy, E. H., Bodnar, L. M., & Naimi, A. I. (2023). Efficient generalization and transportation. arXiv preprint arXiv:2302.00092.
> * Liu, S., Takeda, A., Suzuki, T., & Fukumizu, K. (2017). Trimmed density ratio estimation. Advances in neural information processing systems, 30.
> * Dahabreh, I. J., Robertson, S. E., Steingrimsson, J. A., Stuart, E. A., & Hernan, M. A. (2020). Extending inferences from a randomized trial to a new target population. Statistics in medicine, 39(14), 1999-2014.
> * Pearl, J., & Bareinboim, E. (2022). External validity: From do-calculus to transportability across populations. In Probabilistic and causal inference: The works of Judea Pearl (pp. 451-482).

---

> > ### Author Response · Authors · 2023-08-18
> > **Evidence for assumptions in real data**
> >
> > Our dataset consists of a representative sample of fee-for-service Medicare beneficiaries with more than 100,000 admissions to short-term acute-care hospitals for Acute myocardial infarction (AMI). In this dataset, the positivity assumption of the treatment (A3 and A5) holds for each state because there are hospitals in each state that perform the treatment and there are no baseline covariates that a contraindication for the treatment.
> >
> > The positivity assumption for site selection (A7) is also plausible since none of the states deny admission to AMI patients on the basis of any of the baseline covariates.
> >
> > In the first iteration of data analysis, we also empirically justify the positivity assumptions by calculating state-by-state summary statistics; for example, the proportion of treatment across states is the highest for Wyoming, with a proportion more than 0.45; and is the lowest for Maine, with a proportion around 0.25. We further analyze by sub-cohorts defined by baseline covariates and report no evidence of violations of A3 and A5.
> >
> > The exchangeability assumptions will be evaluated with the proposed sensitivity analysis. Unfortunately, we cannot run the sensitivity analysis at the moment because the authors are physically out of the US and CMS data requires access to the data from the US only.

---

> > ### Comment · Reviewer_LaCg · 2023-08-22
> >
> > Thanks for your rebuttal. I remain concerned about the practical applicability of this work. The authors did not provide convincing evidence for, and unfortunately I did not see an easy path for the work to be practically used in a range of real applications. I keep my score.

---

### Author Rebuttal · Authors · 2023-08-10

* We have responded to each reviewer's comments.
* We also attach a PDF containing results from an updated experiment, in response to Reviewer 27aW's inquiry regarding non-comparable sample sizes in the target site and source sites.

---

### Decision · Program_Chairs · 2023-09-21

**Decision:**

Accept (poster)

**Comment:**

This paper considers federated learning for multi-site studies. The authors propose an estimation approach that is robust to model misspecification and covariate shift, and privacy-preserving. The work synthesizes standard tools from disparate literature and contributes to a narrow setting where the mean exchangeability assumption holds across sites, and covariate shift is the main concern. As the reviewers pointed out, it will be important to provide at least one convincing motivating real application that can be plausibly modeled by the authors' setup. In addition, the authors should better discuss the limitations of their work (e.g., on the privacy side) in the camera-ready version.